# Odor-regulated oviposition behavior in an ecological specialist

Raquel Álvarez-Ocaña [1], Michael P. Shahandeh [1,3], Vijayaditya Ray[2,3], Thomas O. Auer [1], Nicolas Gompel[2] & Richard Benton [1] ✉

Colonization of a novel ecological niche can require, or be driven by, evolution of an animal's behaviors promoting their reproductive success. We investigated the evolution and sensory basis of oviposition in *Drosophila sechellia*, a close relative of *Drosophila melanogaster* that exhibits extreme specialism for *Morinda citrifolia* noni fruit. *D. sechellia* produces fewer eggs than other drosophilids and lays these almost exclusively on noni substrates. We show that visual, textural and social cues do not explain this species-specific preference. By contrast, we find that loss of olfactory input in *D. sechellia*, but not *D. melanogaster*, essentially abolishes egg-laying, suggesting that olfaction gates gustatory-driven noni preference. Noni odors are detected by redundant olfactory pathways, but we discover a role for hexanoic acid and the cognate Ionotropic receptor 75b (Ir75b) in odor-evoked oviposition. Through receptor exchange in *D. melanogaster*, we provide evidence for a causal contribution of odor-tuning changes in Ir75b to the evolution of *D. sechellia*'s oviposition behavior.

Colonization of, and specialization on, a new ecological niche by an animal can provide many benefits, such as access to new resources, protection from biotic and abiotic threats, and avoidance of competition[1]. Niche specialization often requires adaptation of multiple behavioral, physiological and morphological traits to survive and reproduce in a new habitat. Divergence of many traits can potentially lead to reproductive isolation and ultimately speciation, making niche specialization a likely driver of biodiversity[2–4]. Many striking examples of adaptations to new niches are known, from the rapid evolution of beak morphology of Darwin's finches as they radiated across the Galápagos archipelago[5] to visual system loss in Mexican tetra (*Astyanax mexicanus*, blind cave fish) in cave dwellings in the Gulf of Mexico and Rio Grande[6]. While candidate genomic regions and genes have been implicated in some of these adaptations (e.g.,[7,8]), the restricted genetic tractability of these species – and most other examples in nature – limits our understanding of the mechanistic basis of evolutionary adaptations.

The fly *Drosophila sechellia* provides an exceptional model to investigate the genetic and cellular basis of niche adaptation[9–11]. This species is endemic to the Seychelles archipelago, where it has evolved an extreme specialist lifestyle, feeding and breeding exclusively upon the "noni" fruit of the *Morinda citrifolia* shrub. Adaptation to this niche has occurred in the last few 100,000 years, potentially only since its divergence from a last common ancestor with the cosmopolitan generalist, *Drosophila simulans* (Fig. 1a). Importantly, the close phylogenetic proximity of *D. sechellia* to the laboratory model, *Drosophila melanogaster* (Fig. 1a), has facilitated the development of genetic tools in this species to explore the mechanistic basis of niche specialization[11–13]. Previous work identified *D. sechellia* Odorant receptors (Ors) essential for long-range detection of noni odors, Or22a and Or85c/b[13–15], and demonstrated a causal relationship between differences in tuning properties of Or22a in *D. melanogaster* and *D. sechellia* and species-specific noni attraction[13].

Long-range olfactory attraction to noni is only one facet of *D. sechellia*'s phenotypic adaptations in this specialized niche[11]. Notably, the fruit is highly toxic to other drosophilids (and more divergent insects) – predominantly due to its high levels of octanoic

[1]Center for Integrative Genomics, Faculty of Biology and Medicine, University of Lausanne, CH-1015 Lausanne, Switzerland. [2]Evolutionary Ecology, Ludwig-Maximilians Universität München, Fakultät für Biologie, Biozentrum, Grosshaderner Strasse 2, 82152 Planegg-Martinsried, Germany. [3]These authors contributed equally: Michael P. Shahandeh, Vijayaditya Ray. ✉e-mail: Richard.Benton@unil.ch

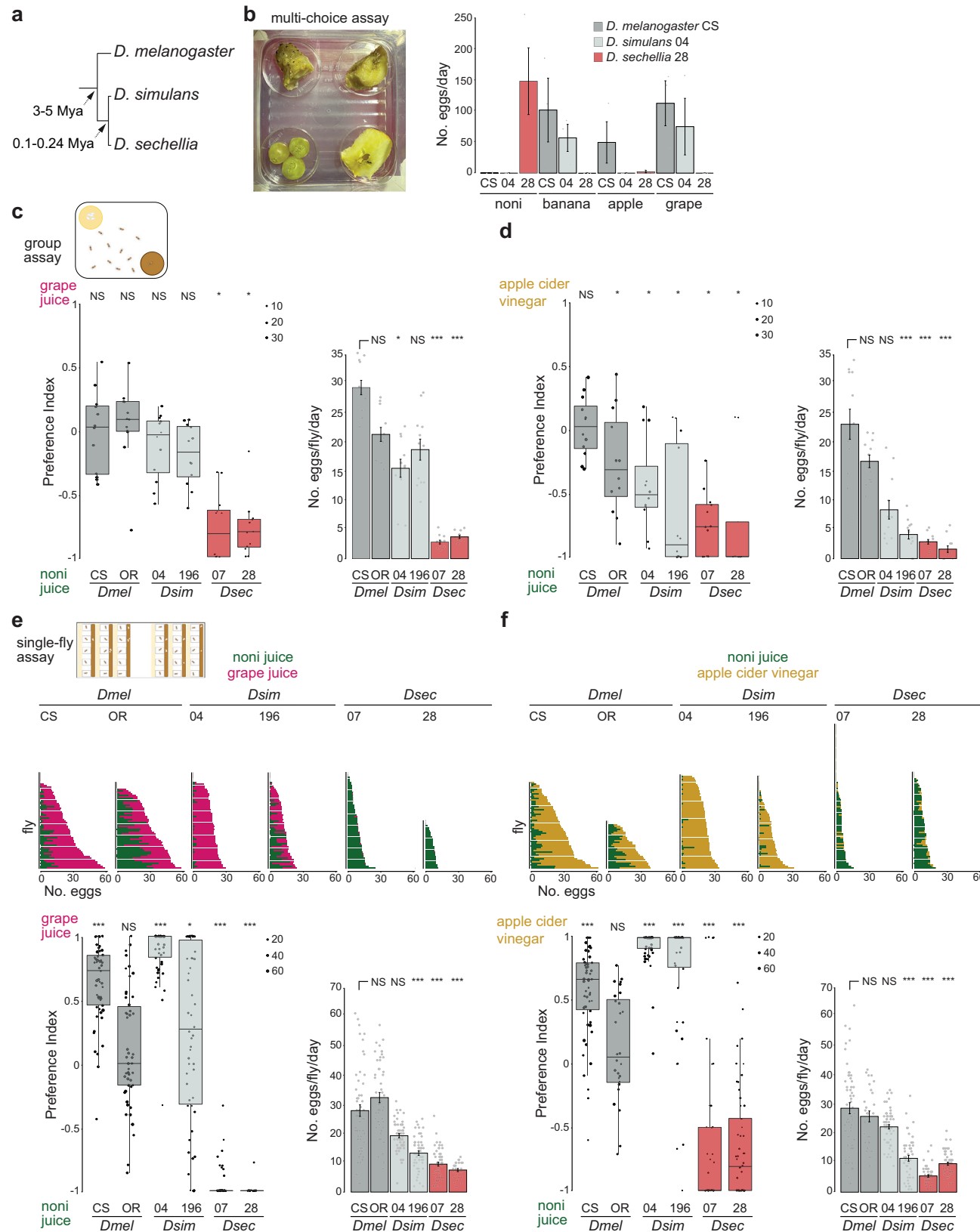

acid – indicating the existence of robust (albeit unclear) resistance mechanisms of *D. sechellia* throughout its life cycle[16–18]. Host fruit toxicity has been suggested to relieve *D. sechellia* from interspecific competition and parasitoidization[19], providing a potential explanation for the selective advantage of its stringent niche specialization.

Another unique set of phenotypes of *D. sechellia* relates to the production and deposition of eggs. Compared to its generalist cousins, *D. sechellia* ovaries contain ~3-fold fewer ovarioles, with a commensurate reduction in egg number[16,20,21]. The evolutionary advantage (if any) of reduced fecundity is unclear, but might be linked with the larger size of *D. sechellia* eggs (~50% by volume[22]) and the greater

**Fig. 1 | *D. sechellia* displays robust, species-specific preference for oviposition on noni substrates. a** Phylogeny of the drosophilid species studied in this work. Mya, million years ago. **b** Fruit multi-choice oviposition preference assay. Left: image of the assay with noni, banana, apple and grape (clockwise from top left) in the arena. Right: quantification of the number of eggs laid per day; *N* = 3 assays/ species, using 50 flies each for a duration of 3 days. Strains used: *D. melanogaster* Canton-S (CS), *D. simulans* 14021-0251.004 (04) and *D. sechellia* 14021-0248.28 (28); see Supplementary Table 1 for details of all strains used in this work. In these and all other bar plots, mean values ± standard error of the mean (SEM) are shown, overlaid with individual data points. All raw behavioral data are provided in the Source Data files. **c** Group oviposition preference assays for noni juice versus grape juice (see Supplementary Table 2 for sources of all chemical stimuli) in 0.67% agarose using two strains each of wild-type *D. melanogaster* (*Dmel*: CS and Oregon R (OR)) (dark grey bars, here and elsewhere), *D. simulans* (*Dsim*: 04 and 14021-0251.196 (196)) (light grey bars, here and elsewhere) and *D. sechellia* (*Dsec*: 14021-0248.07 (07) and 28) (red bars, here and elsewhere). Left: box plots of oviposition preference index. In these and all other box plots, the middle line represents the median, and the lower and upper hinges indicate the first and third quartiles, respectively. Individual data points are overlaid on the box plots; point size is scaled by the total number of eggs laid in an assay (key at top right of the plot); data

beyond the whiskers are considered outliers. For these and other box plots, statistically-significant differences from 0 (no preference) are indicated: ****P* < 0.001; ***P* < 0.01; **P* < 0.05; NS (not significant) *P* > 0.05 (Wilcoxon test with Bonferroni correction for multiple comparisons); *N* = 12 (representing 4 group assays, each scored on 3 successive days with fresh oviposition plates each day). Exact *P* values for the statistical comparisons are provided in the Source Data files. Right: bar plots of egg-laying rate per fly per day in these assays. Statistically-significant differences from the *D. melanogaster* CS strain are indicated: ****P* < 0.001; ***P* < 0.01; **P* < 0.05; NS *P* > 0.05 (Kruskal-Wallis rank sum test with Nemenyi post-hoc test). **d** Group oviposition preference assays, as in **c**, for noni juice versus apple cider vinegar; *N* = 12, as in **c**. **e** Single-fly oviposition preference assays for noni juice versus grape juice in agarose for the same strains as in **c**. Top: total number of eggs laid in each substrate by each female. Bottom left: oviposition preference index. Statistically-significant differences from 0 (no preference) are indicated as in **c**; *N* = 30–60 flies across 1-2 technical replicates (precise *N* values for these and all following assays are provided in the Source Data files). Bottom right: egg-laying rate, presented as in **c**. **f** Single-fly oviposition preference assays, as in **e**, for noni juice versus apple cider vinegar; *N* = 30–90 flies across 1–3 technical replicates.

tendency of this species to retain fertilized eggs (resulting in facultative ovoviviparity)[22]. Such observations hint that these traits could be related to higher investment of *D. sechellia* in fewer eggs to protect them from the acid-rich noni substrate and/or predators. However, non-adaptive explanations (e.g., pleiotropic effects of mutations in genes underlying other adaptations[23,24]) cannot be excluded.

Whatever the reason(s) for reduced fecundity of *D. sechellia*, this trait makes the decision of females to engage in oviposition particularly important. Previous work has shown that *D. sechellia* depends on the presence of noni for both egg production and laying[16,25,26]. Furthermore, the most abundant noni chemicals, hexanoic and octanoic acids, can alone induce oviposition[27,28]. However, the cognate sensory pathways are unknown, as is the contribution (if any) of other chemosensory or non-chemosensory information to this behavior. In this work, we combine diverse behavioral assays with loss- and gain-of-function genetic manipulations in *D. sechellia* and *D. melanogaster* to demonstrate the essential role of olfaction for *D. sechellia* oviposition, and provide evidence for a contribution of the evolved hexanoic acid receptor Ir75b in this species' egg-laying behaviors.

## Results

### Species-specific oviposition preference and rate
To investigate the neurosensory basis of egg-laying behavior in *D. sechellia*, we first compared the specificity of oviposition site selection of *D. sechellia*, *D. simulans* and *D. melanogaster* in a semi-natural, multi-choice assay in which animals were offered slices of different ripe fruits (noni, banana, apple and grape) within an enclosed arena (Fig. 1b). While *D. melanogaster* and *D. simulans* flies avoided using noni fruit as an oviposition substrate (preferring two or three of the other fruits instead), *D. sechellia* laid eggs almost exclusively on noni (Fig. 1b).

To systematically test the species-specificity of oviposition behavior and its sensory basis, we next established two-choice group oviposition assays. In these, egg-laying substrates comprised of commercial juices/vinegar (to ensure consistency of chemical stimuli) mixed in either agarose or Formula 4–24® instant *Drosophila* medium (hereafter, "instant medium"). The latter substrate supported higher egg-laying rate, which was important when examining the influence of single odors in subsequent experiments. Two independent strains of each species were tested in all assays to distinguish interspecific from intraspecific differences. *D. melanogaster* and *D. simulans* strains generally exhibited indifference between noni and grape juice substrates, in either agarose or instant medium, with small differences between strains (Fig. 1c and Supplementary Fig. 1a). By contrast, *D. sechellia* consistently displayed a strong preference for noni

juice-containing substrates (Fig. 1c and Supplementary Fig. 1a). Similar, though less marked, differences between species' preferences were observed in assays offering a choice between noni juice and apple cider vinegar in agarose (Fig. 1d), but not in instant medium (Supplementary Fig. 1b).

As social interactions can influence drosophilids' oviposition preference[29,30], we also established a single-fly oviposition assay (see Methods and Supplementary Fig. 2)[31]. Using the same combinations of stimuli and substrates as in group assays, we observed even more marked species differences in oviposition site preference: *D. sechellia* laid the vast majority of its eggs on noni juice substrates in all assays (Fig. 1e, f and Supplementary Fig. 1c, d), while *D. melanogaster* and *D. simulans* exhibited strong preference for either grape juice or apple cider vinegar in agarose, and variable, strain-specific levels of preference for these counter-stimuli in instant medium.

Both group and single-fly assays also confirmed the substantially lower fecundity of *D. sechellia* compared to *D. melanogaster* and *D. simulans* (Fig. 1c–f and Supplementary Fig. 1). Quantification of eggs laid by individual flies revealed large variation in egg-laying rate for all species, even within a given assay (Fig. 1e, f and Supplementary Fig. 1c, d). However, on average, *D. sechellia* consistently laid a low number of eggs (up to ~5–7 eggs/female/day), while the mean egg-laying frequency for the other species could vary substantially (from ~10 to ~30 eggs/female/day), which might be related to the provision of less or more appealing substrates in different assays.

Together, these results highlight the innate, social context-independent, and robust preference of *D. sechellia* for noni substrates, contrasting with the more context-dependent noni indifference or avoidance exhibited by the generalist drosophilids.

### *D. sechellia* robustly probe the oviposition substrate
In the oviposition experiments with *D. sechellia* on agarose, we observed many small indentations in the substrate surface at the end of the assay (Fig. 2a). Such indentations were only occasionally observed on the agarose substrates where *D. melanogaster* or *D. simulans* had laid eggs (see below). (The presence of indentations could not be easily assessed in the instant medium substrate due to its more granular texture). Furthermore, indentations were not observed on agarose exposed only to *D. sechellia* males, suggesting that they are not the result of non-sexually dimorphic behaviors, such as proboscis probing of the substrate.

The size and shape of the indentations led us to hypothesize that they correspond to the substrate marks formed by the ovipositor during "burrowing" – rhythmic digging of the ovipositor into the substrate

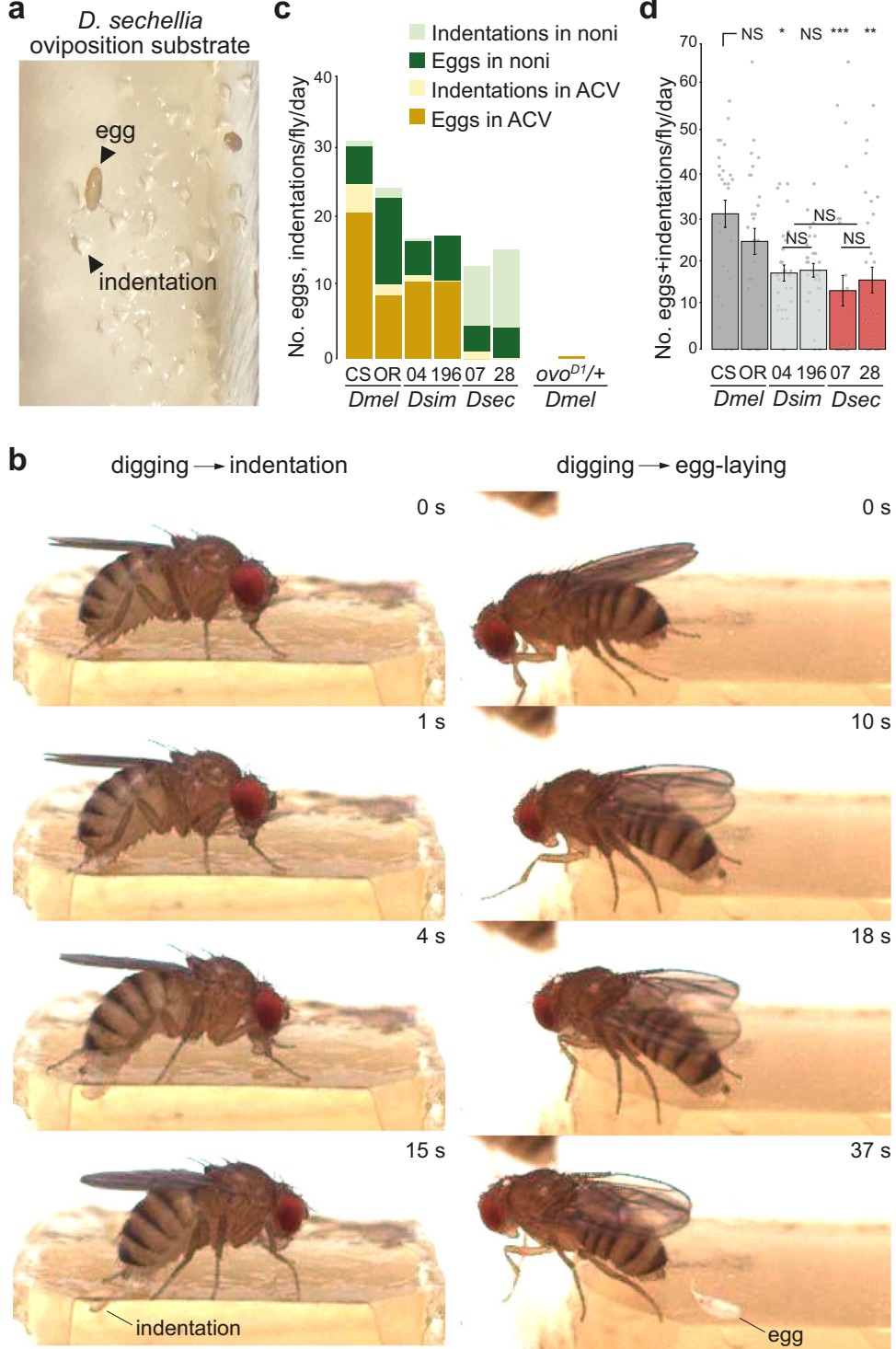

**Fig. 2 | *D. sechellia* make frequent substrate indentations during oviposition.**
**a** Photo of the noni juice/agarose substrate at the end of a single-fly oviposition assay with *D. sechellia* illustrating the many indentations in the agarose surface and rare eggs. **b** Still images from high-speed movie sequences of *D. sechellia* oviposition behavior illustrating a digging event that does not lead to egg deposition, which results in the formation of a visible indentation on the substrate (left), and a digging event that culminates in egg deposition (right). The full movies are provided in Supplementary Movies 4 and 5. **c** Rate and distribution of egg-laying and indentation formation of different species and strains on different substrates in a single-fly two-choice oviposition assay with noni juice and apple cider vinegar (ACV) as oviposition substrates (indicated by different colors in the figure); N = 19–60 flies across 1-2 technical replicates. To obtain *D. melanogaster ovoD1/+* females, *ovoD1v24/Y/C(1)Dx,y,f* males were crossed to *D. melanogaster* CS females. **d** Summed egg-laying and indentation rate of the experiments in **c**. Statistically-significant differences from the CS strain are indicated: \*\*\*$P < 0.001$; \*\*$P < 0.01$; \*$P < 0.05$; NS $P > 0.05$ (Kruskal-Wallis rank sum test with Nemenyi post-hoc test). Exact $P$ values for the statistical comparisons are provided in the Source Data files.

prior to egg deposition – described in *D. melanogaster*[32]. To test this hypothesis, we used high-speed imaging to visualize *D. sechellia* oviposition behavior at high spatio-temporal resolution[33]. Notably, although the total number of egg-laying events captured was low (precluding detailed quantitative analyses), we observed frequent interactions of the ovipositor of these flies with the substrate that did not culminate in egg deposition. These interactions ranged from simple substrate touching or scratching by the ovipositor (Supplementary Movie 1-2; see Methods for classification of behaviors) to more involved digging behaviors (Supplementary Movie 3). In two instances, such digging resulted in the formation of a visible indentation (Fig. 2b and Supplementary Movie 4). However, post-hoc observation of the substrate revealed several other examples of indentations that were not captured during the recordings, possibly because they were not visible from the camera angle to the substrate. Conversely, we did not observe any other behaviors of the fly that could explain the formation of the indentations. Very similar ovipositor-digging events were observed prior to egg laying (Fig. 2b and Supplementary Movie 5).

These observations support the hypothesis that indentations represent aborted oviposition events in *D. sechellia*. We reasoned that they provide a relevant complementary measure of oviposition behavior to the number of eggs laid. We therefore quantified the numbers of both indentations and eggs for all three species in single-fly two-choice assays (Fig. 2c). The total numbers of indentations and eggs were comparable for *D. sechellia* and *D. simulans* strains, and slightly lower than for *D. melanogaster* (Fig. 2d). We also tested *D. melanogaster* mutants that lack eggs (*ovo*[D1]), but found that these flies do not make indentations (Fig. 2c). This observation suggests that the higher rate of indentation formation of *D. sechellia* is not simply a consequence of lower egg number in this species, but rather reflects its robust probing of the oviposition substrate before proceeding to egg deposition.

### *D. sechellia*'s oviposition preference does not require vision

To assess the sensory basis of *D. sechellia*'s strong preference for oviposition on noni substrates, we first tested the contribution of vision, because the natural fruits (as well as the artificial substrates) have characteristic colors that might influence decisions on where to lay eggs. In single-fly two-choice assays run in the dark, *D. sechellia* retained very strong, species-specific, preference for laying on noni juice substrates, and no decrease in egg-laying rate was noted (Fig. 3a).

We extended this analysis to examine whether *D. sechellia* exhibits any unique color preference, reflecting its use of ripe (dull white/yellow) but not unripe (green) fruit. As the noni juice colored the oviposition substrate brown, we tested this possibility using a short-range trap assay[34], in which identical odor traps – containing noni juice for *D. sechellia* or balsamic vinegar for *D. melanogaster* and *D. simulans* – were enclosed within green or white casings (Fig. 3b). Because noni fruit can be found among green foliage or on the white sandy substrate below *Morinda citrifolia* shrubs[11], we reasoned that color contrast might also play an important role in substrate preference, and therefore tested trap preference on a white or green background, as well as in the dark as a control. We observed no preference of any species to enter different colored traps (Fig. 3b), suggesting that color is not a critical cue that *D. sechellia* uses to locate host fruit, at least at short-range.

### *D. sechellia* and *D. simulans* prefer softer substrates

Substrate hardness is another factor influencing oviposition site preference that might have diverged between drosophilid species. For example, the pest species *D. suzukii* – which oviposits in various ripe, but not rotten, fruits – exhibits stronger preference for stiffer substrates (that presumably resemble more closely ripe fruit) than *D. melanogaster*[35]. We compared the texture preference profile of *D. sechellia*, *D. simulans* and *D. melanogaster* through single-fly two-choice assays in which both substrates contain the same attractive chemical stimulus (either noni juice or apple cider vinegar) but

different stiffness, obtained by pairing a soft agarose substrate (0.5%) with one ranging from 0.5–2% agarose. Although all three species preferred to oviposit on softer agarose, the discrimination threshold was different: *D. melanogaster* only exhibited such a preference when 0.5% agarose was paired with 1.25% (or higher) agarose, while *D. simulans* and *D. sechellia* discriminated a more subtle difference in texture, preferring 0.5% agarose over 0.75% agarose (Fig. 3c, d). The textural discrimination ability of *D. sechellia* therefore cannot explain its ecological specialization. However it is consistent with our observations that *D. sechellia* lays its eggs on the softest part of noni fruit: the pedicel cavity in intact fruits (or internal flesh in cut/broken fruits) (Fig. 3e), which is much softer than the fruit skin, whose stiffness is approximately equivalent to 0.75% agarose (Fig. 3f).

### Olfactory pathways required for oviposition behaviors

Having excluded the importance of vision for *D. sechellia*'s egg-laying preference, we reasoned that olfactory cues are likely to be the first sensory signals that *D. sechellia* uses when assessing potential oviposition sites, as these do not require direct contact with the substrate. We first tested near-anosmic double-mutant animals for the conserved olfactory co-receptors Orco (required for the function of all Ors) and Ir8a (required for the function of volatile acid-sensing Irs)[13,36–38]. Strikingly, in single-fly assays – offering a choice of noni juice and apple cider vinegar in agarose – these flies laid essentially no eggs (Fig. 4a, b). Moreover, no indentations were observed on the substrate at the end of the assay (Fig. 4b). This lack of oviposition activity is not due to any overt locomotor defects, as these mutant animals display similar levels of activity as wild-type strains (Supplementary Fig. 3). It is also not due to any decrease in egg production of these mutants, as their ovaries contain a similar number of mature eggs as in wild-type animals (Fig. 4c). Importantly, equivalent *D. melanogaster* near-anosmic *Ir8a;Orco* double-mutant animals lay many eggs (Supplementary Fig. 4). These observations provide evidence that olfactory input is critical for oviposition behavior in *D. sechellia*, but not *D. melanogaster*.

To test whether olfactory cues are sufficient to promote oviposition, we performed a no-choice oviposition assay in which flies were provided with an agarose/sucrose substrate with a non-accessible source of noni juice or, as control, water (Supplementary Fig. 5). No differences were observed in egg-laying rate of *D. sechellia* (or *D. melanogaster*) strains between these two conditions (Supplementary Fig. 5). These results argue that noni odors alone are insufficient to promote egg-laying behavior in *D. sechellia*, which presumably relies also upon gustatory input through multiple contact chemosensory organs, as is the case in *D. melanogaster*[39–41].

To further understand the contribution of olfaction to *D. sechellia*'s oviposition behavior, we next tested the *Orco* and *Ir8a* co-receptor mutants singly: both showed a decreased egg-laying rate compared to wild-type controls, but only *Ir8a* mutants displayed reduced preference for noni juice (Figs. 4a, d). Similar phenotypes for these mutants were observed in two-choice assays with grape juice as a counter-stimulus (Supplementary Fig. 6). We went on to screen the phenotypes of mutants lacking genes encoding individual odor-specific "tuning" Ors and Irs[13], including Or22a and Or85c/b that are required for long-range noni attraction[13]. While these lines displayed variable reductions in egg-laying rate, none of them displayed significantly diminished oviposition preference for noni substrates (Figs. 4a, d and Supplementary Fig. 6) or diminished locomotor activity (Supplementary Fig. 3). The maintenance of robust oviposition preference towards noni in most of these assays suggested that multiple, partially redundant olfactory signals contribute to oviposition behavior in *D. sechellia*.

### Effect of individual noni chemicals on oviposition

To characterize the noni chemicals promoting *D. sechellia*-specific oviposition, we tested several candidates in single-fly assays using the

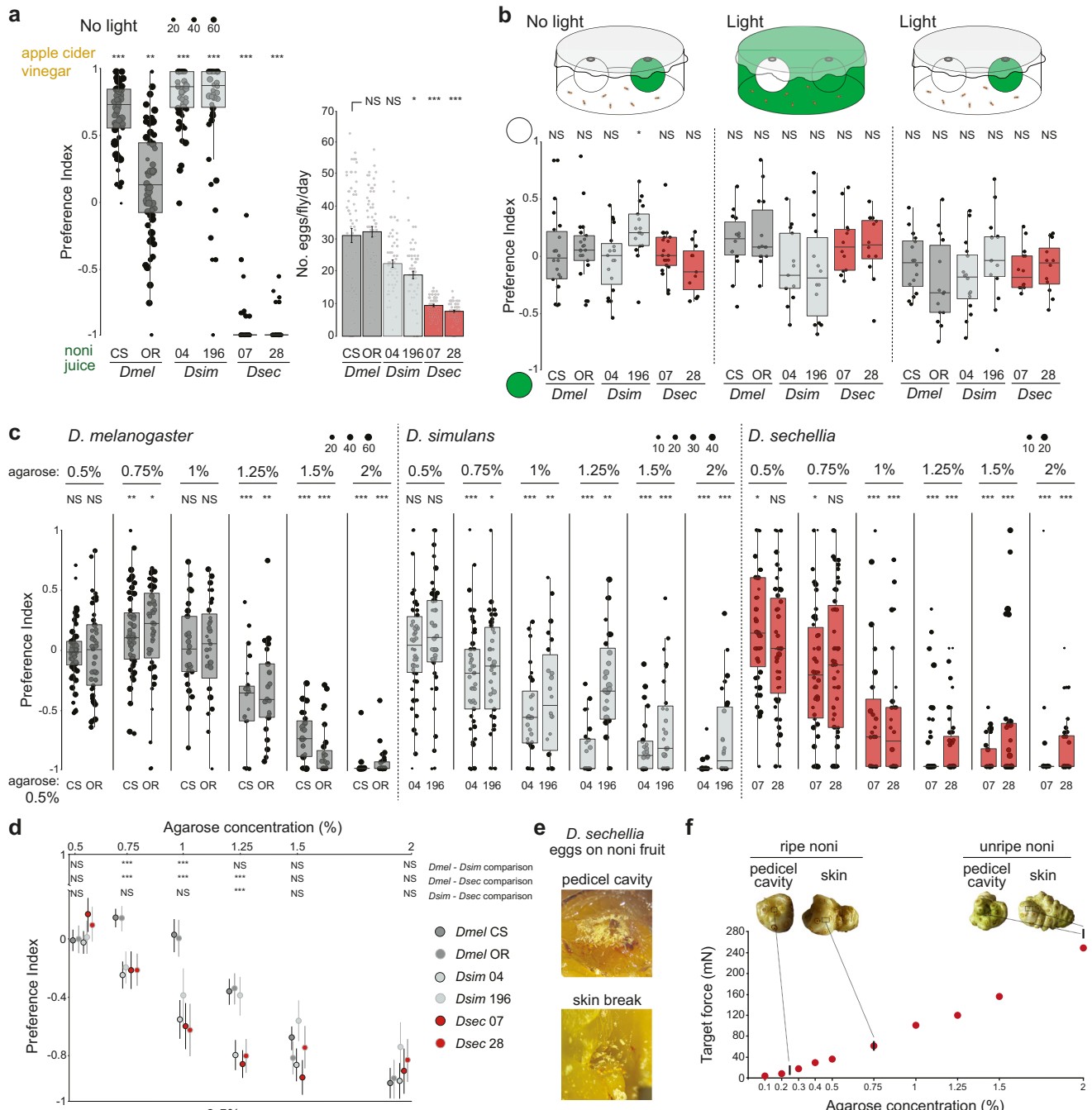

**Fig. 3 | Analysis of visual and textural contributions to *D. sechellia*'s noni preference. a** Single-fly oviposition preference assays in the dark for noni juice versus apple cider vinegar in agarose (fly strains as in Fig. 1c). Left: oviposition preference index. Statistically-significant differences from 0 (no preference) are indicated: ***$P < 0.001$; **$P < 0.01$ (Wilcoxon test with Bonferroni correction for multiple comparisons); $N = 60$ flies across 2 technical replicates. Right: egg-laying rate in these assays; Kruskal-Wallis rank sum test with Nemenyi post-hoc test. Statistically-significant differences from the *D. melanogaster* CS strain are indicated: ***$P < 0.001$; *$P < 0.05$; NS $P > 0.05$. **b** Group color preference assays in which flies are given a choice to enter two traps containing the same chemical stimulus (balsamic vinegar (*D. melanogaster* and *D. simulans*) or noni juice (*D. sechellia*)) and distinguished only by colored casings with different light and background conditions. Statistically-significant differences from 0 (no preference) are indicated: *$P < 0.05$; NS $P > 0.05$ (Wilcoxon test with Bonferroni correction for multiple comparisons); $N = 12–24$ assays across at least 2 technical replicates. **c** Single-fly oviposition preference assays testing between the agarose concentrations indicated at the top and 0.5% agarose in the counter-substrate. Both substrates contain

apple cider vinegar (*D. melanogaster* and *D. simulans*) or noni juice (*D. sechellia*). Statistically-significant differences from 0 (no preference) are indicated: ***$P < 0.001$; **$P < 0.01$ *$P < 0.05$; NS $P > 0.05$ (Wilcoxon test with Bonferroni correction for multiple comparisons); $N = 30–60$ flies across 1-2 technical replicates. **d** Graph recapitulating data from **c**. Dots represent the mean values and the bars represent ± SEM. The statistical comparisons shown are between the most similar strains of the different species: ***$P < 0.001$; NS $P > 0.05$ (Kruskal-Wallis rank sum test with Nemenyi post-hoc test). **e** Close-up image of noni fruit illustrating the concentration of *D. sechellia* eggs in the pedicel cavity (where the fruit was attached to the stem) and in the flesh exposed by a skin break. Flies were placed in a group assay oviposition chamber containing whole noni fruits during 72 h. **f** Graph of stiffness of substrates of different agarose concentrations (in noni juice), overlaid with the stiffness ranges of unripe and ripe noni fruits (illustrated in the photos) within the pedicel cavity or on the external skin. Measurements were made using Semmes-Weinstein Monofilaments following the procedure described in[78]. Exact $P$ values for the statistical comparisons are provided in the Source Data files.

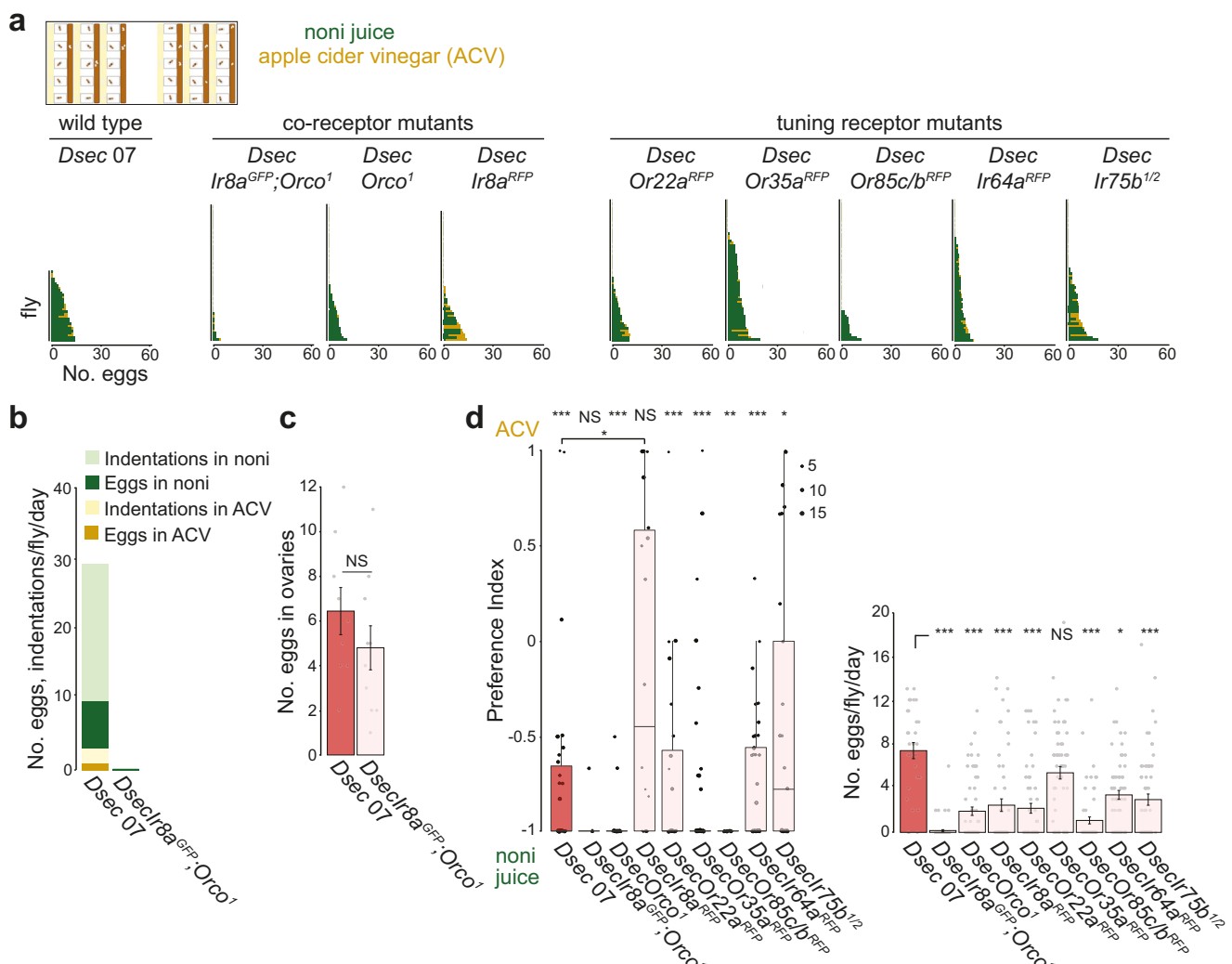

**Fig. 4 | Olfactory pathways required for *D. sechellia* oviposition. a** Single-fly oviposition preference assays for noni juice versus apple cider vinegar in agarose for the indicated genotypes (Supplementary Table 1). The plots show the number of eggs laid per fly; $N = 30$–60 flies across 1-2 technical replicates. *DsecIr75b^{1/2}* is a transheterozygous mutant combination. **b** Quantification of the number of eggs and indentations on different substrates of the indicated genotypes; $N = 30$ (*Dsec* 07) and 58 (*DsecIr8a^{GFP};Orco^1*) across 1-2 technical replicates. **c** Mean number ($\pm$ SEM) of mature eggs per fly (i.e., a pair of ovaries) of the indicated genotypes. NS $P > 0.05$ (two-sample t-test); $N = 9$–10 flies. **d** Left: oviposition preference index for the assays shown in **a**. Statistically-significant differences from 0 (no preference)

are indicated: ***$P < 0.001$; **$P < 0.01$; *$P < 0.05$; NS $P > 0.05$ (Wilcoxon test with Bonferroni correction for multiple comparisons); $N = 30$–60 flies across 1-2 technical replicates. *Dsec* 07 and *DsecIr8a^{RFP}* are statistically-significantly different: $P = 0.0328$ (Wilcoxon test with Bonferroni correction). Right: egg-laying rate. Statistically-significant differences from the *Dsec* 07 strain are indicated: ***$P < 0.001$; *$P < 0.05$; NS $P > 0.05$ (Kruskal-Wallis rank sum test with Nemenyi post-hoc test). The non-significant preference index for the *DsecIr8a^{GFP},Orco^1* double mutant was calculated from the 4/60 animals that laid >2 eggs. Exact $P$ values for the statistical comparisons are provided in the Source Data files.

different species and three odor concentrations (0.05%, 0.1%, 0.5%) (Fig. 5 and Supplementary Fig. 7). Confirming and extending previous group assays[27,28,42], we found that hexanoic acid promoted very strong preference in *D. sechellia* at all concentrations tested (Fig. 5a) and a higher egg-laying rate (compared to water-only control substrates) at least at intermediate concentrations (Fig. 5b). *D. melanogaster* and *D. simulans* both displayed slight preference or indifference at lower concentrations of hexanoic acid and strong aversion at the highest concentration (Fig. 5a), but egg-laying rate was not greatly influenced by this chemical (Fig. 5b). Octanoic acid has also been described to be an oviposition stimulant/attractant for *D. sechellia* in some[27,42,43], though not all[22], reports. In our assays, this acid did not evoke strong oviposition preference of *D. sechellia* at lower concentrations; moreover, egg laying was largely suppressed at the highest concentration, although the very few eggs laid were found on the octanoic acid

substrate (Fig. 5a, b). *D. melanogaster* and *D. simulans* generally found this odor aversive (Fig. 5a).

We tested two other noni chemicals that are behaviorally-important for long-range noni location: methyl hexanoate (detected by Or22a) and 2-heptanone (detected by Or85c/b)[13–15]. *D. sechellia* did not display a strong oviposition site preference for substrates containing methyl hexanoate, although this chemical stimulated a slight enhancement of egg laying at intermediate concentrations (Fig. 5a, b). Similarly, neither *D. melanogaster* nor *D. simulans* exhibited strong preference or aversion to methyl hexanoate-containing substrates (Fig. 5a). 2-heptanone had little consistent influence on oviposition-site selection of any species (Fig. 5a, b); we were unable to use the highest concentration (0.5%) as this was toxic for flies.

Lastly, we tested oviposition stimulants described in *D. melanogaster*, valencene and limonene, which are detected by Or19a

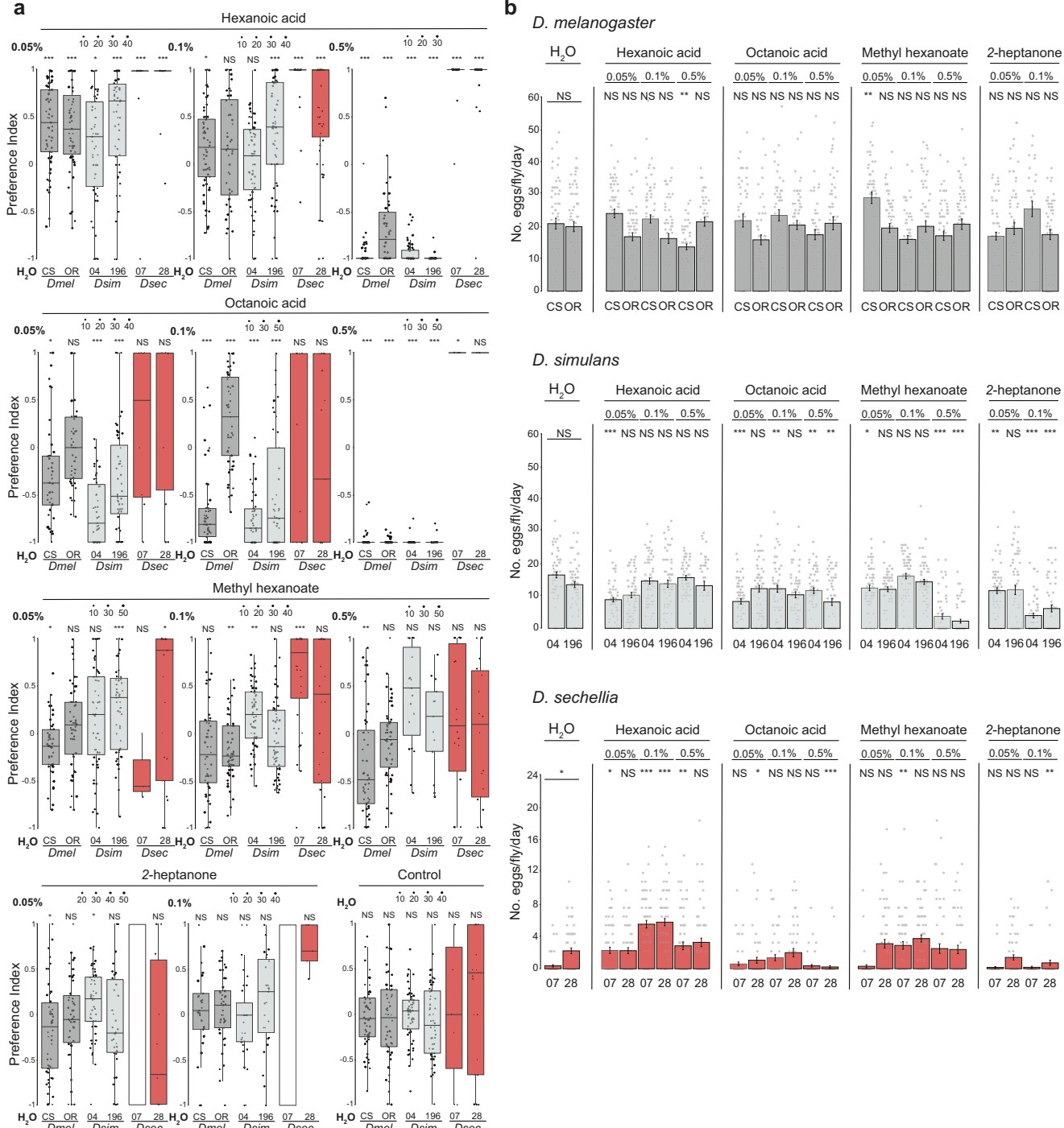

**Fig. 5 | Analysis of the effect of individual noni chemicals on oviposition. a** Single-fly oviposition assays of the indicated strains testing different odors and concentrations in an instant medium substrate. Oviposition preference index. Statistically-significant differences from 0 (no preference) are indicated: ***$P < 0.001$; **$P < 0.01$; *$P < 0.05$; NS $P > 0.05$ (Wilcoxon test with Bonferroni correction for multiple comparisons); $N = 30$–60 flies across 1-2 technical replicates.

For *Dsec* 07 assays with *2*-heptanone (indicated with a white rectangle), the low number of flies laying eggs prevented calculation of a preference index. **b** Egg-laying rate of the assays in **a**. Statistical comparisons of the effect of odors on egg-laying rate were performed across strains: ***$P < 0.001$; **$P < 0.01$; *$P < 0.05$; NS $P > 0.05$ (Kruskal-Wallis rank sum test with Nemenyi post-hoc test). Exact $P$ values for the statistical comparisons are provided in the Source Data files.

neurons[44]. *D. sechellia* flies were indifferent to, or avoided oviposition on, substrates containing either of these chemicals, and egg laying was suppressed at high stimulus concentrations (Supplementary Fig. 7). Interpretation of these results must be tempered, however, with our inability to consistently reproduce behavioral effects of these compounds on oviposition reported in *D. melanogaster* (Supplementary

Fig. 7)[44], potentially reflecting observations that the behavioral function of this olfactory pathway is context-dependent[45].

Together, these experiments reveal the complex, concentration-dependent influence of different individual chemicals on drosophilid oviposition behavior, which might be due to their detection via both olfactory and gustatory systems. The robust and *D. sechellia*-specific

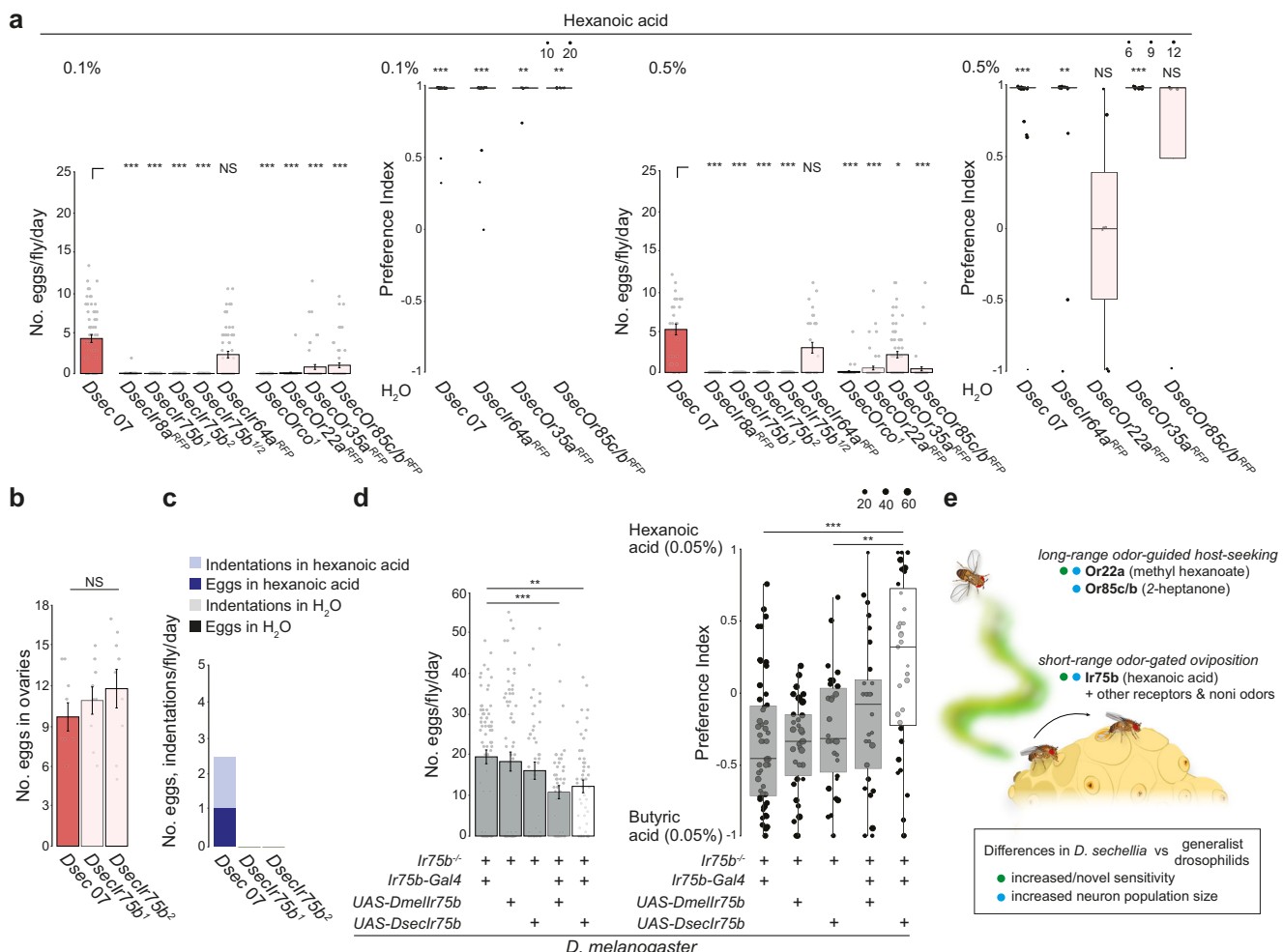

**Fig. 6 | *D. sechellia* Ir75b is required for hexanoic acid responses and sufficient to shift oviposition preference in *D. melanogaster*. a** Single-fly oviposition assays testing $H_2O$ versus 0.1% hexanoic acid and $H_2O$ versus 0.5% hexanoic acid in instant medium. Left: egg-laying rate. \*\*\*$P < 0.001$; \*$P < 0.05$; NS $P > 0.05$ (Kruskal-Wallis rank sum test with Nemenyi post-hoc test). Right: oviposition preference indices are only shown for genotypes that laid at least 2 eggs in these assays. Statistically-significant differences from 0 (no preference) are indicated: \*\*\*$P < 0.001$; \*\*$P < 0.01$; NS $P > 0.05$ (Wilcoxon test with Bonferroni correction for multiple comparisons); $N = 30$–83 flies across 1–3 technical replicates. **b** Mean number (± SEM) of mature eggs per pair of ovaries per fly. NS $P > 0.05$ (two-sample t-test); $N = 9$ flies. **c** Quantification of the number of eggs and indentations on different substrates of the indicated genotypes; $N = 30$ flies in 1 technical replicate. **d** Single-fly oviposition assays testing 0.05% hexanoic acid versus 0.05% butyric acid of *D. melanogaster* *Ir75b* mutant and rescue genotypes. Left: egg-laying rate for *Ir75b-Gal4* control

(*w;Ir75b-Gal4/+;Ir75b^{DsRed}*), *UAS-DmelIr75b* control (*w;UAS-DmelIr75b/+;Ir75b^{DsRed}*), *UAS-DsecIr75b* control (*w;UAS-DsecIr75b/+;Ir75b^{DsRed}*), *DmelIr75b* rescue (*w;Ir75b-Gal4/UAS-DmelIr75b;Ir75b^{DsRed}*), and *DsecIr75b* rescue (*w;Ir75b-Gal4/UAS-DsecIr75b;Ir75b^{DsRed}*). Both rescue strains showed small but significant reduction in the number of eggs compared to *DmelIr75b-Gal4* control. Right: oviposition preference indices for these genotypes. No significant differences were detected between *Ir75b-Gal4* control, *UAS-DmelIr75b* control and *DmelIr75b* rescue strains. The *DsecIr75b* rescue strain showed a significant shift in preference toward 0.05% hexanoic acid compared to both *DmelIr75b-Gal4* and *UAS-DsecIr75b* controls. \*\*\*$P < 0.001$; \*\*$P < 0.01$ (Wilcoxon tests with Bonferroni correction for multiple comparisons); $N = 28$–55 flies per genotype measured across at least 2 technical replicates. **e** Schematic summarizing the contributions of different olfactory pathways to niche specialization in *D. sechellia*. Exact $P$ values for the statistical comparisons are provided in the Source Data files.

effect of hexanoic acid on oviposition led us to focus on determining the sensory mechanism by which this noni chemical is detected.

### *D. sechellia*'s hexanoic acid sensor Ir75b promotes egg laying

To define the sensory mechanisms of volatile hexanoic acid-mediated control of oviposition behavior, we tested our panel of *D. sechellia* olfactory receptor mutants. Loss of either Ir8a or Orco alone led to abolished or greatly diminished egg laying on hexanoic acid substrates (Fig. 6a), suggesting that both Ir and Or pathways contribute. Within the Ir repertoire, Ir75b was an excellent candidate as this receptor has evolved novel sensitivity to hexanoic acid in *D. sechellia* from the ancestral butyric acid sensitivity of the *D. melanogaster* and *D. simulans* orthologs[34]. Indeed, mutation of *Ir75b* in *D. sechellia* led to complete loss of egg laying on hexanoic acid substrates, a phenotype confirmed in two independent alleles, and a transheterozygous *Ir75b* mutant

combination (Fig. 6a). Dissection of these flies' ovaries revealed a similar number of eggs as in controls (Fig. 6b), suggesting the defect was in egg laying not production. Consistent with this hypothesis, *DsecIr75b* mutant flies also produced no indentations in these assays (Fig. 6c). By contrast, *D. sechellia* lacking the broadly-tuned volatile acid sensor, Ir64a[46], still oviposited, laying almost all eggs on hexanoic acid substrates (Fig. 6a).

Amongst the Ors, all three mutants (*DsecOr22a*, *DsecOr35a* and *DsecOr85c/b*) displayed reduced egg-laying rate (Fig. 6a). Of the subset of flies that did lay eggs, the *DsecOr35a* and *DsecOr85c/b* mutants maintained strong preference for oviposition on hexanoic acid substrates while *DsecOr22a* mutants no longer discriminated this substrate from the control medium (Fig. 6a). Or22a neurons are generally considered to be ester sensors in drosophilids[47], but weak Or22a-dependent hexanoic responses have been described in

*D. sechellia*[13] (as well as in *D. melanogaster*[48], where it is the most sensitive hexanoic acid sensor of this species[49]), suggesting that it might be a second olfactory pathway for this oviposition stimulant (see Discussion).

## Evolution of Ir75b contributes to species-specific behavior

Given the important role of Ir75b for hexanoic acid-stimulated oviposition of *D. sechellia*, we asked if the evolution of the tuning of this receptor might explain species-specific oviposition behavior. In an *Ir75b* mutant of *D. melanogaster*[50], we rescued Ir75b function through transgenic expression of either *D. sechellia* Ir75b or, as a control, *D. melanogaster* Ir75b. Confirming our previous analysis of these transgenes in other neuron classes[34,51], we validated their expression and functional differences in Ir75b neurons (Supplementary Fig. 8a, b). These flies were offered a choice to lay eggs on substrates containing hexanoic acid or butyric acid, the preferential ligand of the receptors from *D. sechellia* and *D. melanogaster*, respectively[34]. Egg-laying rate was broadly comparable between all control mutant and both rescue lines (Fig. 6d). This result indicates that a functional Ir75b pathway is not important for egg laying in *D. melanogaster* – consistent with our observation that near-anosmic *D. melanogaster* lay many eggs – thereby permitting assessment of the contribution of the Ir75b pathway to oviposition preference. Rescue flies expressing *D. melanogaster* Ir75b displayed a substrate preference that was not significantly different from parental genotypes (Fig. 6d). By contrast, expression of *D. sechellia* Ir75b was sufficient to shift oviposition preference from butyric acid to hexanoic acid substrates compared to controls (Fig. 6d). These results support a causal contribution of *Ir75b* to the evolution of oviposition site preference during *D. sechellia*'s host specialization. We note that wild-type *D. sechellia* displays a much stronger preference for hexanoic acid over butyric acid (Supplementary Fig. 8c); it is possible that the higher number of Ir75b neurons in *D. sechellia* than in *D. melanogaster* (and *D. simulans*)[34], as well as other sensory adaptations (e.g., in gustatory inputs), contribute to this species' acid preferences.

## Discussion

Decisions on when and where to lay an egg are critical for all oviparous animals to maximize the chance of survival of their offspring, in particular those lacking parental care[52,53]. As such, these decisions are influenced by multiple biotic and abiotic factors in the environment. When species establish themselves within a new ecological niche, changes in these factors can exert selective pressures for novel or modified behavioral responses to sensory cues. It is also possible that chance evolution of traits can permit exploitation of a new niche. Either way, studying differences between species can provide insight into the relative importance of the plethora of environmental signals, as well as the mechanisms by which nervous systems evolve, changing the relationship between these cues and behavioral outputs. *D. sechellia* offers an excellent opportunity to study oviposition behavioral adaptations, both because its specialist lifestyle likely constrains the set of pertinent sensory signals and because its low fecundity presumably renders the decision to lay an individual egg more important than for highly fertile species. Moreover, the phylogenetic proximity of *D. sechellia* to the generalists *D. melanogaster* and *D. simulans* facilitates comparative behavioral and genetic analyses that might enable reconstruction of the (still-unknown) evolutionary history of this species[11,54].

Studies in *D. melanogaster* and other drosophilids have revealed that oviposition decisions are complex, multisensory-dependent behaviors[35,52,55,56], guided both by substrate properties and the recent experience of animals[57]; moreover, the choice of egg-laying site is often assay-dependent (e.g.,[58,59]). Using several types of behavioral assays, we have confirmed the importance of noni for *D. sechellia* for egg-laying rate and site selection. The latter trait contrasts with the

variable preferences of *D. melanogaster* and *D. simulans*. We also discovered an unappreciated feature of oviposition behavior of *D. sechellia*: extensive probing of the substrate surface, resulting in the formation of numerous indentations. These indentations are most likely equivalent to the "burrows" resulting from aborted oviposition events of *D. melanogaster*[32]. One explanation for the high rate of indentations in *D. sechellia* is that females engage in the initiation of the oviposition routine unaware of the low number of eggs they carry. This seems unlikely, however, as it would represent a futile energetic investment for these animals, and *D. melanogaster* mutants that lack eggs do not make indentations. Moreover, high-resolution behavioral observations suggest that the presence of the egg in the ovipositor is integral to penetration of the substrate in *D. melanogaster*[32] and *D. sechellia*. We favor a hypothesis that extensive indentation formation by *D. sechellia* reflects greater choosiness of this species to deposit eggs only after the female has ascertained to have found the optimal substrate available.

To account for the species-specificity of *D. sechellia* substrate selection, we have been able to exclude several sources of sensory information. Visual input is unimportant (at least at short-range), and *D. sechellia* does not exhibit obvious changes in preference for colors that mimic the choice this species makes in nature. While *D. sechellia* prefers to lay eggs within the softest part of the fruit, and within softer agar, there is no difference in texture preference compared to *D. simulans*, suggesting that this trait is not a key facet of host adaptation, contrasting with the fresh fruit feeder *D. suzukii*[35]. Finally, although communal egg laying is widespread in many invertebrates and vertebrates[60], we do not find evidence that this phenomenon contributes to noni preference. If anything, isolated flies lay more eggs with stricter noni preference than those in groups, possibly because they are less distracted by social interactions.

Our genetic analysis indicates that olfactory input is essential for egg laying in *D. sechellia*, as near-anosmic flies fail to lay eggs even in the presence of noni, despite normal egg production. Conversely, exposure of flies to noni odors alone, without allowing them to have gustatory sensation of noni juice, does not enhance oviposition rate. Together, these observations suggest that both olfactory and gustatory inputs are important: without olfaction, gustatory signals are insufficient for promoting oviposition, but olfactory signals without gustatory inputs are similarly ineffective. A future priority is to determine how *D. sechellia* detects noni via gustation and if, as in the olfactory system, any such gustatory pathways differ between drosophilids.

While loss of the vast majority of olfactory input prevents egg laying on noni in *D. sechellia*, we found substantial redundancy, as mutation of any single tuning Or or Ir (or even Orco) did not strongly diminish noni preference. This observation indicates that multiple distinct odors, acting via several different olfactory receptors, must contribute to short-range behavioral decisions. By simplifying the noni odorscape in our oviposition assays we demonstrate the unique oviposition-promoting role of hexanoic acid and, importantly, define Ir75b and its obligate co-receptor Ir8a, as the cognate sensory receptor. Although hexanoic acid might also be detected by gustatory neurons (based upon studies in *D. melanogaster*[61–63]), the selective expression of Ir75b and Ir8a in the antenna argues that this is an odor-guided behavior. Moreover, the demonstration that Ir75b is required for this behavior provides an explanation for the evolutionary changes described in this sensory pathway: while *D. melanogaster* and *D. simulans* Ir75b are tuned primarily to butyric acid, the *D. sechellia* receptor has evolved novel sensitivity to hexanoic acid, through amino acid substitutions within the ligand-binding domain[34,51]. In addition, *D. sechellia* exhibits a 2-3-fold increase in number of sensory neurons expressing Ir75b, resulting in increased sensory pooling onto partner interneurons in the brain[34]. Importantly, replacement of *D. melanogaster* Ir75b with the *D. sechellia* receptor induces a small but significant shift in oviposition site preference, indicating that receptor

tuning changes are sufficient alone to confer more *D. sechellia*-like behavior on *D. melanogaster*.

Together with previous work[13], our studies of noni-dependent odor-guided behaviors in *D. sechellia* reveal similarities and differences in the sensory coding properties and evolution of olfactory pathways mediating long-range and short-range host detection (Fig. 6e). The high redundancy in short-range olfactory signals contrasts markedly with olfactory contributions to long-range noni seeking, where loss of single tuning receptors essentially abolishes the ability of flies to locate the odor source[13]. This difference might reflect the complexity of the noni odor blend at different spatial scales: there are likely to be fewer, highly-volatile, compounds reaching behaviorally-relevant concentrations at a distance compared to odors present at short-range[13]. Concordantly, the most-important receptors for long-range (Or22a and Or85c/b) and short-range (Ir75b) noni detection, display differences in sensitivity: Ir75b neurons require several orders of magnitude higher odor concentrations to evoke the same level of neuronal firing as Or22a or Or85c/b neurons[13]. The segregation of behavioral function of the pathways is, however, not absolute: the long-range olfactory detectors, notably Or22a, also appear to contribute to oviposition behaviors on hexanoic acid substrates, though further genetic analysis will be necessary to confirm this.

One striking commonality of all three of these OSN populations is their expansion in *D. sechellia*, although the functional significance of this phenotype is unknown. By contrast, the nature of odor specificity evolution of these pathways is different: Or85c/b neuron sensitivity to *2*-heptanone is unchanged across the drosophilid species[13], *D. sechellia* Or22a has enhanced sensitivity to methyl hexanoate compared to the orthologous receptor in *D. melanogaster* (but not in *D. simulans*)[13,14], while Ir75b has acquired new sensitivity to hexanoic acid specifically in *D. sechellia*[34,51]. In addition, methyl hexanoate and *2*-heptanone, but not hexanoic acid, are emitted by a wide range of fruits[64]. A model to unify these observations is that methyl hexanoate and *2*-heptanone act as "habitat odor cues"[65], attracting *D. sechellia* (but also other species) in the vicinity of noni, while hexanoic acid is a specific "host odor cue"[65] that, through Ir75b, evokes short-range behaviors only in *D. sechellia*.

In this context, the ecological role of the Ir75b sensory pathway in *D. melanogaster* is unclear, although optogenetic activation experiments have provided evidence for a role in positional attraction and oviposition preference[34,66]. Several other olfactory pathways have been implicated in oviposition promotion in *D. melanogaster*[52,67,68], including Or19a, which detects the citrus odors valencene and limonene[44]. Interestingly, *D. sechellia* Or19a neurons appear to have lost sensitivity to these odors[44], which are not reliably detected in noni fruit[13]. Adaptation of this species might therefore have involved sensory gain or loss in several olfactory pathways to match the pertinent chemical signals in its niche.

Finally, beyond the issues mentioned above, a key future question – in any species – is how olfactory input controls oviposition behavior. Recent studies in *D. melanogaster* have defined circuitry linking mating and egg laying[69]; notably, the activity of some of the component neuron populations (i.e., oviENs and oviINs in the central brain) are activated or inhibited by gustatory and mechanosensory inputs[69]. It is possible that olfactory sensory pathways (e.g., downstream of the Ir75b sensory neurons) impinge on this circuitry[70]. Alternatively, olfactory signals might have only an indirect influence, for example, by modulating gustatory inputs to this egg-laying circuitry. Further exploration of the neural basis of oviposition in *D. sechellia* should yield insights into the mechanistic basis of the adaptations of this critical behavior to this species' unique lifestyle.

## Methods
### Drosophila culture
*Drosophila* stocks were cultured in a 25 °C incubator under a 12 h light:12 h dark cycle on a wheat flour–yeast–fruit juice food. Unless noted otherwise, *D. sechellia* culture vials were supplemented with noni paste, consisting of a few grams of Formula 4–24® instant *Drosophila* medium, blue (Carolina Biological Supply) and noni juice (Raab Vitalfood Bio). All strains used in this study are listed in Supplementary Table 1 and sources of chemicals are listed in Supplementary Table 2.

### Oviposition assays
We maximized flies' egg-laying capacity by following the protocol of[31]: in brief, prior to the experiments, ~50 1-2 day-old females and males were collected and placed in new fly food tubes enriched with dry yeast (*D. melanogaster* and *D. simulans*) or with dry yeast and noni paste (*D. sechellia*) for 5 days. At this point the food was typically full of crawling larvae, inducing females to retain eggs until they were transferred to the assay chamber. Unless otherwise stated, oviposition assays were performed at 25 °C, 60% relative humidity and a 12 h light:12 h dark cycle (starting assays in the early afternoon), in either an incubator or behavior room for 22–72 h (depending upon the assay, see below).

Previous work suggested that the low egg-laying rate of *D. sechellia* is due to alterations in dopamine metabolism – which contributes, at least indirectly, to fertility in *D. melanogaster*[71,72] – and could be partially compensated by supplementation of food with the dopamine precursor 3,4-dihydroxyphenylalanine (L-DOPA), which is found in noni fruit[25]. To increase *D. sechellia*'s oviposition rate, we cultivated flies' for five days on noni food supplemented with L-DOPA (1 mg/ml), but did not observe increased egg laying in either group or single-fly assays compared to control flies given only noni paste (Supplementary Fig. 9a). Treatment with α-methyl-DOPA (0.4 mM), a non-hydrolyzable L-DOPA analog that acts as a competitive inhibitor of DOPA decarboxylase (which converts L-DOPA to dopamine) reduced egg laying in the single-fly assay but not the group assay (Supplementary Fig. 9a). Our inability to fully reproduce the reported effects on oviposition[25] – we did not examine other traits investigated in that study, such as egg size and germline cyst apoptosis – might be due to experimental differences in our assays (e.g., use of noni fruit) or the use of more fertile *D. sechellia* strains.

We also tested whether substrates containing freshly-extracted noni juice substrates – obtained by crushing ripe noni fruits, removing the seeds, centrifuging in a 15 ml Falcon at 4000 rcf for 10 min and collecting the supernatant ("noni extract") – might be more attractive than those prepared with commercial noni juice; we did not find that fresh juice induced higher oviposition rates, nor any differences in indentation formation (Supplementary Fig. 9b, c, d).

**Fruit multiple-choice assay.** Ripe fruits (apple, banana, grape (all from Migros); noni from *M. citrifolia* plants (University of Zurich Botanical Gardens and Canarius) grown in a greenhouse) were cut into thick (1-2 cm) slices. Fruit pieces were placed in a 5 cm diameter Petri dish (Falcon) inside a plastic chamber (15 cm length × 14 cm width × 5 cm height; Migros). For all species, 50 females and 20 males were anesthetized on ice and introduced into the chamber, which was covered with a fabric gauze. The chambers were placed in a behavioral room in constant darkness for 72 h, after which the number of eggs on each fruit piece was quantified.

**Group two-choice assay.** Agarose (Promega) substrates were prepared as follows: a 1% agarose solution was prepared and let to cool until it was possible to hold the glass Erlenmeyer flask with bare hands. The 1% agarose preparation was added to the juice/odor solution in a 2:1 ratio, resulting in a final concentration of 0.67% agarose. The final mixture was poured up to a 0.5 cm depth into 3 cm diameter Petri dishes (Falcon). Agarose plates were kept at 4 °C for a maximum of 3 days. Instant medium substrates were prepared by diluting 12 g of instant medium in 100 ml of noni juice (or apple cider vinegar or grape juice) creating a semi-solid consistency. The instant medium was spread in a 3 cm diameter Petri dish until fully covering the bottom of

the plate. Instant medium mixes were kept at 4 °C and used within two weeks. Two Petri dishes containing the desired combination of substrates were placed into the same chamber used for the fruit multiple-choice assay. 10 females and 8–10 males (*D. melanogaster* and *D. simulans*) or 20 females and 15–20 males (*D. sechellia*) were anesthetized with $CO_2$ and introduced into the chamber for 3 days. Due to *D. sechellia*'s low fecundity, through preliminary experiments we considered that the number of eggs laid by 20 *D. sechellia* female flies was sufficient to observe a clear behavioral preference between two conditions. Petri dishes were exchanged with fresh ones every 24 h by quickly lifting the mesh cover. The number of eggs laid per substrate was counted independently on each plate.

**Group no-choice odor cue assay.** As for the "*Group two-choice assay*" except using a single 3 cm diameter Petri dish containing 0.67% agarose and 150 mM sucrose (Sigma) onto which a non-accessible container covered with a fabric gauze and a perforated cap of a 15 ml Falcon tube (diameter 1 cm, height 2 cm; Techno Plastic Products AG) into which 300 µl of $H_2O$ or noni juice was pipetted.

**Single-fly two-choice assay.** 30-cell single-fly chambers were designed and manufactured by Formoplast S.A. following published blueprints[31], but using poly(methyl methacrylate) instead of acrylic, and adding a small handle to the top door. Flies were anesthetized in $CO_2$ and placed in individual egg-laying chambers. Animals were allowed a 30 min period for recovery from anesthetization and acclimation to the chamber, during which time the oviposition substrate was prepared as described above. For single-odor assays, instant medium was dissolved in the desired odor solution diluted in water or juice. The concentration range of odors was defined from preliminary tests and previous studies[27]; specified concentrations represent those before adding the instant medium. For agarose substrates, 1 ml of agarose solution containing the stimuli was added to the appropriate wells of the chamber (each underlying one side of 5 separate cells) to produce the desired combination of substrates within each cell. For instant medium substrates, the paste was added to the wells with a spatula. The fly loading compartment was placed on top of these wells to allow flies access to the substrates. Eggs were scored on each substrate after 22 h. Preliminary tests of oviposition preferences on different substrates, which informed subsequent experimental design, are shown in Supplementary Fig. 2.

**Texture assays.** Single-fly oviposition assays were performed as described above but preparing substrates with different final concentrations of agarose.

For all assays, eggs were scored manually under a binocular microscope. The oviposition preference index was calculated as: (number of eggs on substrate X - number of eggs on substrate Y)/total number of eggs. The preference index was not calculated for assays where <2 eggs were laid. Indentations were scored as small breaks/holes in the substrate surface; in some cases, the presence of multiple indentations in the same region of the substrate likely led to underestimation of the number of independent indentations.

**High-speed imaging of oviposition behavior**
*D. sechellia* flies were cultured on cornmeal media. Mated *D. sechellia* females (aged 4–10 days in cornmeal food vials with or without supplementation with noni paste), were placed in empty vials with hydrated tissue for 14–20 h before the day of the experiment to force egg retention. A group of 2-3 individuals was introduced into a cubical oviposition filming chamber[33]. A trough on one side of the chamber was filled with a noni juice-agar substrate (1:3 or 1:6 v/v), with a wet pad at its base to limit sagging caused by desiccation. Flies were filmed with a high-speed camera (JAI RMC-6740 GE, IMACO as detailed in[33]) using the custom FlyBehavior software[33] (https://github.com/LMU-AgGompel/FlyBehaviour), focused on the surface of the agar column, which provided a 1 × 3 mm oviposition substrate. Recording of the group was performed for several separate 2 h-sessions over the course of one day (from late morning to mid-evening). The following specific behaviors were observed qualitatively by visual inspection of the resulting Movies: *Touching* (Supplementary Movie 1) - the ovipositor simply touches the substrate; *Scratching* (Supplementary Movie 2) - the ovipositor brushes against the substrate, giving the impression of gentle scratching; *Digging* (Supplementary Movie 3) - the ovipositor burrows into the substrate surface; *Indentation formation* (Supplementary Movie 4) - the fly digs with its ovipositor at a particular site on the substrate leaving a minor depression (indentation) but no egg; occasionally, we observed that a fly returns to an indentation for egg laying; *Egg laying* (Supplementary Movie 5) - the fly starts digging into the surface and lays an egg at this site. In many movies, we also observed flies exuding a liquid droplet, possibly from their anal plates (e.g., Supplementary Movie 1, example 4, left-hand animal); this action does not appear to be related to oviposition as it was observed also in virgin female and male flies. Movie sequences were cropped and assembled in Fiji[73].

**Color preference assays**
Color preference was assessed by adapting an olfactory trap assay[34], in which the arena contained two traps filled with 300 µl of the same attractive odor – noni juice (*D. sechellia*) and balsamic vinegar (Antica Modena) (*D. melanogaster* and *D. simulans*) – masked with different visual cues. To simulate fruit at a ripe or unripe stage, a trap was covered with a green or white matte table-tennis ball (Lakikey; 40 mm diameter) with two opposing holes cut into it: one large hole on the bottom to insert the trap vial, and a smaller hole on top to allow the flies to enter the trap through a 200 µl pipette tip that was flush with the ball surface. Arenas were lined with either green or white paper (to provide different contrast for the green and white traps), and assays were performed in the light as well as under complete darkness in a behavior room (25 °C and 60% relative humidity). Prior to the assay, flies were kept on wheat flour-yeast-fruit juice media without noni supplement for 24 h. Twenty-five fed and mated 3–5 day-old females were introduced into each arena after brief ice anesthesia. The number of flies in each trap (as well as untrapped animals) was counted after 24 h; replicates where >25% of flies died within the experimental period were discarded. The preference index was calculated as: (number of flies in white trap - number of flies in green trap)/number living flies (trapped and untrapped).

**Locomotor activity monitoring**
Activity was measured for 5–7 day old mated females at 25 °C under a 12 h light: 12 h dark cycle, staged as for oviposition assays to ensure mating status, in the *Drosophila* activity monitor (DAM) system[74] in incubators with continuous monitoring of light and temperature conditions (TriTech Research DT2-CIRC-TK). In brief, this system uses an infrared beam that bisects a 5 mm glass tube, in which the fly is housed, to record activity as the number of beam crossings per minute. Each tube is plugged with a 5% sucrose/2% agar (w/v) food source at one end and cotton wool at the other. Each DAM was used to record the activity of up to 32 flies simultaneously, and multiple monitors were contained in a single incubator. For all genotypes, we recorded flies over at least 2 technical replicates. Mean activity of an animal was calculated as the average number of beam crossings per minute over 3 complete days of recording.

**Ovary dissection and egg quantification**
Females, prepared as for the oviposition assays, were anesthetized with $CO_2$ and their ovaries dissected with forceps in phosphate buffered saline, using a surgical needle to separate the ovarioles. Mature eggs present in each ovary were counted under a binocular microscope.

## Histology

Immunostainings on whole-mount antennae were performed following a published protocol[75] using guinea pig α-Ir75b (RRID: AB_2631093)[76] (diluted 1:100) and Alexa488 α-guinea pig (A11073 INVITROGEN AG) (diluted 1:500). Images were acquired using ZEN 2.3 SP1 software on a Zeiss LSM 710 confocal microscope and processed using Fiji[73].

## Electrophysiology

Single sensillum electrophysiological recordings from female flies were performed following published protocols[77] using chemicals of the highest purity purchased from Sigma Aldrich. Odorants (butyric acid (CAS 107-92-6) and hexanoic acid (CAS 1821-02-9)) were used at 1:100 (v/v) dilutions in solvent (double-distilled water) for a maximum of 5 consecutive trials. Corrected responses were calculated as the number of spikes in a 0.5 s window at stimulus delivery (200 ms after stimulus onset to take account of the delay due to the air path for olfactory stimulation) subtracting the number of spontaneous spikes in a 0.5 s window 2 s before stimulation, multiplied by 2 to obtain spikes/s. The solvent-corrected responses shown in the figures were calculated by subtracting from the response to each diluted odor the response obtained when stimulating with the solvent. Recordings were performed on a maximum of 3 sensilla per fly.

## Statistical analyses

Oviposition preference indices were calculated compared to the null hypothesis (i.e., preference index = 0) for each strain using a Wilcoxon test with Bonferroni correction for multiple comparisons. In all box plots, the middle line represents the median, and the lower and upper hinges indicate the first and third quartiles, respectively; whiskers extending out of the box indicate the variability beyond the upper and lower quartiles; data points beyond the whiskers are considered outliers. All bar plots of egg-laying rate show the mean ± SEM. Statistically-significant differences across the number of eggs laid per fly per day for multiple comparisons were calculated applying Kruskal-Wallis rank sum test with Nemenyi post-hoc test. For two-sample comparisons, a two-sample t-test was used. The reference strain for multiple or two-sample comparisons is indicated in the figure legends. Error bars show SEM. All statistical values reported on the figures are as follows: NS (not significant) $P > 0.05$; $*P < 0.05$; $**P < 0.01$; $***P < 0.001$. As egg-laying rate of *D. sechellia* sometimes displayed quantitative variation between experiments performed at different times, statistical comparisons were only performed between assays performed in parallel. All behavioral data were analyzed using Microsoft Excel and statistical analyses were performed in R.

## Reporting summary

Further information on research design is available in the Nature Portfolio Reporting Summary linked to this article.

## Data availability

All materials and data supporting the findings of this study are available from the corresponding author on request. Source data from behavioral and electrophysiological experiments, together with statistical analyses, are provided with this paper.

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

## Acknowledgements

We thank Chung-Hui Yang for valuable assistance in establishing the single-fly oviposition assay, Stefano Ceolin for advice on video recordings, René Gerber (University of Zurich Botanical Gardens) for a gift of *M. citrifolia*, Blaise Tissot-Dit-Sanfin for maintenance of *M. citrifolia*, Daniel Pauli for *D. melanogaster* stocks, Steeve Cruchet and Liliane Abuin for technical support, and members of the Benton laboratory for discussions and comments on the manuscript. This work was supported by the Graduate School of Systemic Neurosciences (V.R.), a Deutsche Forschungsgemeinschaft grant (GO 2495/9-1) (N.G.), a Human Frontier Science Program Long-Term Fellowship (LT000461/2015-L) and a Swiss National Science Foundation Ambizione Grant (PZ00P3 185743) (T.O.A.), and an ERC Advanced Grant (833548) and the Swiss National Science Foundation (R.B.).

## Author contributions

R.A.-O. and R.B. conceived the project. R.A.-O. performed the majority of experiments. M.P.S. performed the experiments in Figs. 3b and 6d and Supplementary Fig. 3; V.R., supervised by N.G., performed the experiments in Fig. 2b and Supplementary Movies 1–5. T.O.A. generated *D. sechellia* lines, performed the electrophysiology experiments in Supplementary Fig. 8a, b, and contributed to the establishment of the assays in Figs. 1b and 3e. All authors contributed to experimental design and interpretation of results. R.B. and R.A.-O. wrote the manuscript with contributions from all authors. All authors read and approved the final manuscript.

## Competing interests

The authors declare no competing interests.
