## [Peer Review File · Nature Communications]

Odor-regulated oviposition behavior in an ecological specialistReviewer #1 (Remarks to the Author):

The authors report experiments on *Drosophila sechellia*, in comparison with *Drosophila melanogaster* and partially *Drosophila simulans*, to reveal which sensory signals guide *D. sechellia*'s highly selective preference for oviposition on noni fruits. The work contributes to a better understanding of evolutionary mechanisms underlying ecological niche formation.

The specific findings are:

1. *D. sechellia* shows a preference for noni fruit juice as the substrate for oviposition.
2. *D. sechellia* probes the substrate before oviposition, similar to other *Drosophila* species, and *D. sechellia* prefers softer substrates than *D. melanogaster*.
3. Visual cues are dispensable for oviposition site choice.
4. Oviposition site choice depends on the chemical cue hexanoic acid, and on its detection through the ionotropic receptor *Ir75b*. Evolutionary "tuning" of this receptor is suggested to guide the species-specific egg-laying behavior.

Some of these findings confirm previous published results. However, the identification of *Ir75b* as the most important receptor for selecting the oviposition site is novel and highly interesting.

The experiments are clearly described and illustrated, the statistical analysis is sound, the methodology is appropriate and according to standards in the field, and the text is excellently written and well understandable. I could not detect any flaw in the data analysis, interpretation or conclusion. Overall, the study is highly interesting for a broad readership, and I recommend publication in *Nature Communications*.

I have two suggestions:

1. Whereas it is a beautiful finding that species-specific properties of *Ir75b* contribute to the species' ecological niche, it would be informative to directly compare amino acid sequences of *Ir75b* between the *Drosophila* species investigated here. After all, what is termed "evolutionary tuning" here is actually mutation. What exactly has been "tuned" through mutation in the course of evolution?

2. The toxicity of noni fruits for most *Drosophila* and other insect species is mentioned in the introduction (lines 108-115). However, the interesting finding by Lavista-Llanos (2014) that high L-DOPA levels in noni fruits are toxic is mentioned only in the Methods section. Since the publication by Lavista-Llanos also suggests an evolutionary adaptation that contributes to the formation of the specific ecological niche, it would be nice to discuss it.

Best regards,
André Fiala

Reviewer #2 (Remarks to the Author):

In this MS, Benton and colleagues examined how oviposition of *Drosophila sechellia* is regulated by olfaction. Since *D. sechellia* is a known "specialist" species that exclusively oviposits on the noni fruits, comparing how oviposition is differentially regulated in *D. sechellia* vs. a generalist such as *D. melanogaster* offers a unique opportunity for understanding the genetic, cellular, and circuit basis of niche adaptation, an important research area. To that end, the authors reported several interesting differences between the two species. First, *sechellia* but not *melanogaster* strictly requires olfactory input for oviposition. Second, *sechellia* and *melanogaster* have different preferences for chemical content and hardness of oviposition substrates. Third, *sechellia* has much lower rate of

egg-laying than melanogaster, and fourth, Ir75a from the two species differ in their sensitivity to hexanoic acid, an important chemical in the noni fruit that promotes oviposition in sechellia. Overall, I find the experiments well executed and the results clear and well put together. But ultimately I am unenthusiastic about this paper because it is mostly descriptive and offers no genuine conceptual advances. It reports several interesting features of sechellia oviposition that distinguish them from melanogaster but, barring one interesting but somewhat problematic Ir rescue experiment, it does not truly address what causes these two species to differ in their niche specialization: i.e., why sechellia is a specialist and melanogaster a generalist.

Major issue

The most interesting experiment in this MS was the one that assessed whether sechellia's strong preference for hexanoic acid over that of melanogaster may be partly explained by the differences in their Ir75a receptor. The authors claimed it is the case but I am not fully convinced. I suggest the following experiments to further confirm this important conclusion. First, compare the preference of WT melanogaster and sechellia in butyric acid vs. hexanoic acid two-choice assay. Second, test if Ir75a mutant melanogaster reduce their preference for hexanoic acid in a hexanoic acid vs. water two-choice assay (as opposed to butyric acid vs. hexanoic acid). Third, test if melanogaster mutants rescued with Ir75a from these two species behave differently in the hexanoic acid vs. water two-choice assay. Fourth, test by e-phys if Ir75 mutant melanogaster rescued with Ir75a from the two species show different odor response to hexanoic acid. Results from these experiments will provide a clearer understanding about the role of Ir75a and Ir75a neurons in melanogaster oviposition in response to hexanoic acid as well as whether enhanced sensitivity of these neurons conferred by sechellia Ir75a alone may push melanogaster to favor hexanoic acid more (like sechellia does).

Minor issue

1) It might be worth mentioning in the discussion and/or introduction that one important feature of melanogaster oviposition is that, aside from its lack of clear host specificity, the decision on whether to deposit an egg on a substrate is clearly influenced by recent experience (as opposed to purely by the quality of the substrate currently being explored, an approach likely adopted by sechellia.) See a recent Science Advance paper "An internal expectation guides *Drosophila* egg-laying decisions" from the Maimon lab.

2) I am a little bit confused about the oviposition rate in response to hexanoic acid reported in Figure 5 and 6. In Figure 5B, sechellia oviposition nearly halved when hexanoic acid increased from 0.1% to 0.5%. But in Figure 6A, Dsec07 laid similar if not more eggs on 0.5% as compared to on 0.1%.

3) Line 549: the correct names for these neurons are oviEN and oviIN.

NCOMMS-22-39514-T: RESPONSE TO REVIEWERS

We thank the reviewers for their careful reading and constructive criticisms of our manuscript. Below, we provide responses to each of the raised issues.

Referee 1

The authors report experiments on *Drosophila sechellia*, in comparison with *Drosophila melanogaster* and partially *Drosophila simulans*, to reveal which sensory signals guide *D. sechellia*'s highly selective preference for oviposition on noni fruits. The work contributes to a better understanding of evolutionary mechanisms underlying ecological niche formation.

The specific findings are:

1. *D. sechellia* shows a preference for noni fruit juice as the substrate for oviposition.
2. *D. sechellia* probes the substrate before oviposition, similar to other *Drosophila* species, and *D. sechellia* prefers softer substrates than *D. melanogaster*.
3. Visual cues are dispensable for oviposition site choice.
4. Oviposition site choice depends on the chemical cue hexanoic acid, and on its detection through the ionotropic receptor *Ir75b*. Evolutionary "tuning" of this receptor is suggested to guide the species-specific egg-laying behavior.

Some of these findings confirm previous published results. However, the identification of *Ir75b* as the most important receptor for selecting the oviposition site is novel and highly interesting.

The experiments are clearly described and illustrated, the statistical analysis is sound, the methodology is appropriate and according to standards in the field, and the text is excellently written and well understandable. I could not detect any flaw in the data analysis, interpretation or conclusion. Overall, the study is highly interesting for a broad readership, and I recommend publication in *Nature Communications*.

I have two suggestions:

1. Whereas it is a beautiful finding that species-specific properties of *Ir75b* contribute to the species' ecological niche, it would be informative to directly compare amino acid sequences of *Ir75b* between the *Drosophila* species investigated here. After all, what is termed "evolutionary tuning" here is actually mutation. What exactly has been "tuned" through mutation in the course of evolution?

RESPONSE: In fact, as cited in our manuscript, we have previously characterised the molecular evolution of the tuning profile of *D. sechellia* *Ir75b* (Prieto-Godino *et al.*, *Neuron* 2017 and Prieto-Godino *et al.*, *eLife* 2021): in brief,

we demonstrated that a single amino acid substitution in the ligand-binding pocket of *D. melanogaster* Ir75b (to the residue present in the *D. sechellia* receptor) is sufficient to confer novel sensitivity to hexanoic acid, but additional pocket mutations, as well as some on the external surface of the ligand-binding domain help to refine the specificity. We have improved our phrasing in the revised manuscript to better highlight this foundational work. Furthermore, as these previous studies expressed Ir75b variants in the “Ir decoder” neuron (i.e., Ir84a neurons lacking Ir84a), in this revision we have now analysed the expression and functional properties of *D. melanogaster* and *D. sechellia* Ir75b when heterologously expressed in *Ir75b* mutant neurons (Supplementary Figure S8), confirming their differences in specificity.

Importantly, neither of these earlier studies examined the behavioral role of this pathway in *D. sechellia* or the significance of the changes in tuning of Ir75b, which is what we accomplish in the present work.

2. The toxicity of noni fruits for most *Drosophila* and other insect species is mentioned in the introduction (lines 108-115). However, the interesting finding by Lavista-Llanos (2014) that high L-DOPA levels in noni fruits are toxic is mentioned only in the Methods section. Since the publication by Lavista-Llanos also suggests an evolutionary adaptation that contributes to the formation of the specific ecological niche, it would be nice to discuss it.

RESPONSE: There appears to be a slight misunderstanding: the work of Lavista-Llanos *et al.*, eLife 2014 provided evidence that L-DOPA is required for efficient egg-laying of *D. sechellia* rather than being the toxic component of this fruit. (The toxic compounds, notably octanoic acid, were suggested to preserve L-DOPA levels by preventing oxidation). Inspired by this work, we made some efforts to replicate the influence of L-DOPA provision on oviposition, in part for practical reasons to help obtain more eggs for oviposition preference assays. However, we were unable to robustly reproduce this effect. As such, we decided it would be simplest to describe these (largely negative) results in the Methods section, so that our replication attempt was reported, while avoiding disrupting the logical flow of the results.

Referee 2

In this MS, Benton and colleagues examined how oviposition of *Drosophila sechellia* is regulated by olfaction. Since *D. sechellia* is a known “specialist” species that exclusively oviposits on the noni fruits, comparing how oviposition is differentially regulated in *D. sechellia* vs. a generalist such as *D. melanogaster* offers a unique opportunity for understanding the genetic, cellular, and circuit basis of niche adaptation, an important research area. To that end, the authors reported several interesting differences between the two species. First, *sechellia* but not *melanogaster* strictly requires olfactory input for oviposition. Second, *sechellia* and *melanogaster* have different preferences for chemical content and hardness of oviposition substrates. Third, *sechellia* has much lower rate of egg-

laying than *melanogaster*, and fourth, Ir75b from the two species differ in their sensitivity to hexanoic acid, an important chemical in the noni fruit that promotes oviposition in *sechellia*. Overall, I find the experiments well executed and the results clear and well put together. But ultimately I am unenthusiastic about this paper because it is mostly descriptive and offers no genuine conceptual advances. It reports several interesting features of *sechellia* oviposition that distinguish them from *melanogaster* but, barring one interesting but somewhat problematic Ir rescue experiment, it does not truly address what causes these two species to differ in their niche specialization: i.e., why *sechellia* is a specialist and *melanogaster* a generalist.

RESPONSE: We agree with the reviewer that why *D. sechellia* is a specialist and *D. melanogaster* a generalist is a fundamental question, but there is no singular (or necessarily simple) answer to this question. Niche adaptation inevitably involves a suite of changes in the phenotype of species (e.g., behavioral, physiological, morphological). The motivation of our study was to understand the nature and basis for changes in one critical behavior in *D. sechellia* – oviposition – and, in this context, the work has yielded several important advances:

- setting a standard for evolutionary neurobiology: multi-strain, multi-assay investigation, incorporating critical genetic tools in a non-traditional model system, and cross-species gene replacement to test causality for phenotypic differences
- first demonstration of an absolute requirement for olfaction for gating oviposition behavior in insects
- first demonstration of a role for an olfactory Ir (Ir75b) in oviposition in insects
- evidence relating changes in modification in odor tuning of this Ir to species-specific behavioral preference (we address specific comments regarding this experiment in more detail below)
- genetic separation of the olfactory requirements for long-range and short-range olfactory behaviors in *D. sechellia*
- identification of high rates of substrate probing (i.e., “indentation” formation) as a potential novel behavior in *D. sechellia*

We do not claim that changes in oviposition behavior in *D. sechellia* were the initial drivers of noni specialization, and we were not expecting to be able to make such a claim at the start of this work. Modifications in long-range odor-guided host-seeking and gustatory behaviors have also been described in *D. sechellia* (e.g., Auer *et al.*, Nature 2020; Reisenman *et al.*, 2023), building up a complex picture of how this species has adapted to its niche. Ultimately, defining the order of adaptations of a species is incredibly hard; describing the nature of individual phenotypic changes is an essential prerequisite to be able to reconstruct an evolutionary model of the selective advantage(s) of niche specialization (i.e., “why” a species is a specialist).

Major issue

The most interesting experiment in this MS was the one that assessed whether *sechellia*' strong preference for hexanoic acid over that of *melanogaster* may be partly explained by the differences in their Ir75b receptor. The authors claimed it is the case but I am not fully convinced. I suggest the following experiments to further confirm this important conclusion.

RESPONSE: We respond below to each of the suggestions; we assume the referee is referring to Ir75b (not Ir75a) in their comments. For clarity, in this document, we have taken the liberty to edit the referee's comments to replace "Ir75a" with "Ir75b" (except when they refer to both pathways separately)

First, compare the preference of WT *melanogaster* and *sechellia* in butyric acid vs. hexanoic acid two-choice assay.

RESPONSE: We now provide these data (for two different wild-type strains for both species) in Supplementary Figure 8c. As expected, *D. sechellia* displays strong preference for oviposition on a hexanoic acid substrate, while *D. melanogaster* does not display an obvious preference. It is difficult to directly compare these preferences with those of the mutant/transgenic flies shown in Figure 6d, because the genetic background of the latter is very different, which is why we limited our statistical comparisons to this set of genotypes (which were tested in parallel).

Although we see a shift in preference towards the hexanoic acid substrate in the *D. melanogaster* strain expressing *D. sechellia* Ir75b, the preference is clearly much lower than that of wild-type *D. sechellia*. This is unsurprising to us, as these genotypes differ phenotypically in many ways. For example, *D. sechellia* has 2-3-fold more Ir75b neurons than *D. melanogaster* (Prieto-Godino *et al.*, Neuron 2017), a neuronal phenotype not yet possible to reproduce by genetic manipulation of *D. melanogaster*. Moreover, it is very likely that gustatory cues from these substrates will also contribute to the species-specific preference (Reisenman *et al.*, J Exp Biol 2023). Nevertheless, the fact that we do observe a shift provides evidence for a role for the change in tuning of Ir75b to modification of oviposition site preference of these species.

Second, test if Ir75b mutant *melanogaster* reduce their preference for hexanoic acid in a hexanoic acid vs. water two-choice assay (as opposed to butyric acid vs. hexanoic acid).

Third, test if *melanogaster* mutants rescued with Ir75b from these two species behave differently in the hexanoic acid vs. water two-choice assay.

RESPONSE: We respond to the two comments above collectively as they have been addressed with the same set of experiments, shown in **Reviewer Figure 1** (see the end of this document). We already showed the oviposition responses of wild-type strains of *D. melanogaster* to hexanoic acid vs control (water)

substrates in Figure 5A, revealing a dose-dependent response from slight attraction (0.05% hexanoic acid), neutrality (0.1%) to strong aversion (0.5%), which dramatically contrasts with the robust attraction of *D. sechellia* to hexanoic acid substrates at all three concentrations. As described in the response above, because the genetic background of the wild-type *D. melanogaster* strains is different from those carrying the mutant allele and multiple different transgenes, we restricted our direct comparisons of preferences to the *Ir75b* mutant background (three different genotypes) and the two rescue genotypes (either with *D. melanogaster Ir75b* or *D. sechellia Ir75b*). In brief, we observed no difference in oviposition preference between any mutant or rescue genotype for any of the three concentrations of hexanoic acid tested. (We also note that qualitative comparison of these responses to those of the wild-type strains shown in Figure 5 indicate a global shift in the dosage sensitivity, i.e., in all mutant/rescue lines the lowest hexanoic acid concentration leads to neutral preference, while both higher concentrations lead to avoidance.)

Our interpretation of these results is that behavioral responses of *D. melanogaster* to hexanoic acid in this assay are probably independent of endogenous *Ir75b* (which was expected as this receptor does not respond to hexanoic acid in this species), and that other sensory inputs (e.g., gustatory sensing of hexanoic acid) drive the behavioral avoidance such that it overrides any hexanoic acid-dependent olfactory activation of *Ir75b* neurons in the *D. sechellia Ir75b* rescue genotype. Our two-choice assay (Figure 6d) – with low concentrations of hexanoic acid and butyric acid – circumvents the innate aversion of *D. melanogaster* to hexanoic acid, and allow us to assess the behavioral differences between the *D. melanogaster* and *D. sechellia Ir75b* rescue genotypes. We re-iterate that the evolution in tuning of *Ir75b* is only one of the sensory changes contributing to the species-specific oviposition preference, and it is inevitable that detecting this contribution behaviorally necessitates a sensitive assay design.

Fourth, test by e-phys if *Ir75b* mutant *melanogaster* rescued with *Ir75b* from the two species show different odor response to hexanoic acid.

RESPONSE: We previously demonstrated the functional differences between *D. melanogaster* and *D. sechellia Ir75b* through electrophysiological analysis of these receptors' tuning profiles when expressed in the *Ir* decoder neuron (i.e., *Ir84a* neurons lacking *Ir84a*) (Prieto-Godino *et al.*, Neuron 2017 [Figure 3D] and Prieto-Godino *et al.*, eLife 2021 [Figure 5A]). To further support the interpretation of the behavioral experiments presented in the current manuscript, we have now analyzed the expression and functional properties of *D. melanogaster* and *D. sechellia Ir75b* when heterologously expressed in *Ir75b* mutant neurons (Supplementary Figure S8a-b). In brief, we find the receptor proteins are expressed at comparable level in *Ir75b* neurons, but – as expected – while *D. melanogaster Ir75b* responds strongly to butyric acid, but not hexanoic acid, *D. sechellia Ir75b* confers strong response to hexanoic acids and weaker responses to butyric acid.

Results from these experiments will provide a clearer understanding about the role of Ir75a and Ir75b neurons in melanogaster oviposition in response to hexanoic acid as well as whether enhanced sensitivity of these neurons conferred by sechellia Ir75b alone may push melanogaster to favor hexanoic acid more (like sechellia does).

RESPONSE: We hope to have assuaged the reviewer's concerns with our responses above regarding the contribution of tuning changes in *D. sechellia* Ir75b to the changes in oviposition preference of this species.

We are not sure why the reviewer makes reference to "Ir75a neurons" in *D. melanogaster*, as neither these (nor the receptor Ir75a) were the subject of study in this work. It is possible some confusion may have arisen because initial RNA FISH analysis (Benton *et al.*, Cell 2009) indicated that Ir75a and Ir75b are co-expressed, but subsequent work indicated that the proteins are expressed in distinct neuronal populations (Prieto-Godino *et al.*, Neuron 2017). The RNA "co-expression" arises from run-off transcription from the *Ir75b* gene into the neighboring *Ir75a* gene, i.e., exons of the latter, excluding exon 1, are incorporated into the 3'-UTR of *Ir75b* transcripts but do not encode Ir75a protein (Mika *et al.*, Science Advances 2021).

Minor issue

1) It might be worth mentioning in the discussion and/or introduction that one important feature of melanogaster oviposition is that, aside from its lack of clear host specificity, the decision on whether to deposit an egg on a substrate is clearly influenced by recent experience (as opposed to purely by the quality of the substrate currently being explored, an approach likely adopted by sechellia.) See a recent Science Advance paper "An internal expectation guides Drosophila egg-laying decisions" from the Maimon lab.

RESPONSE: We read this paper with interest while our work was under review; while much further experimentation would be required to determine whether *D. sechellia* exhibits the same kind of expectation behavior as shown for *D. melanogaster* (the focus of that study), we now cite this paper in the Discussion to emphasize the complex nature of the information that guides oviposition decisions.

2) I am a little bit confused about the oviposition rate in response to hexanoic acid reported in Figure 5 and 6. In Figure 5B, sechellia oviposition nearly halved when hexanoic acid increased from 0.1% to 0.5%. But in Figure 6A, Dsec07 laid similar if not more eggs on 0.5% as compared to on 0.1%.

RESPONSE: We suspect that this difference mostly reflects the variation in egg-laying rate of *D. sechellia* (and other drosophilid species) that we observe between experiments performed at different times, where many difficult-to-control

technical factors (precise cultures conditions, seasonal variation in animal state etc.) might influence egg-laying. This variation led us to limit our statistical comparison to experiments performed in parallel. However, taking together the experiments from Figure 5 and Figure 6 that the reviewer points out, we believe that the global conclusion is sound: *Dsec 07* lays very few eggs on a water-only substrate (<0.5/fly/day) (Figure 5) and this rate is increased, when hexanoic acid is included (up to 2-5 eggs/fly/day) (Figure 5 and 6A), even though there is quantitative variation between these different experiments. We profit to point out that many other studies examining oviposition preference in drosophilids do not report egg-laying rate, but only the proportion of eggs laid on different substrates, possibly because of the variability in laying rate (e.g., Dweck *et al.*, Curr Biol 2013; Chen & Amrein Curr Biol 2017; Chen *et al.*, PNAS 2019). We wanted to be explicit in our work about both egg-laying rate and preference, going as far as to scale the data point sizes for preference indices in our plots by the number of eggs laid, reasoning that we can be more confident about the preference of an individual animal that has laid dozens of eggs rather just 2-3 eggs in total.

3) Line 549: the correct names for these neurons are oviEN and oviIN.

RESPONSE: These names have been corrected.

Beyond the new experiments requested by the reviewers described above, we note the following further additions to the manuscript:

1. Having established a ready supply of fresh noni fruit in our laboratory, we were interested to compare *D. sechellia*'s oviposition behavior towards substrates containing fresh fruit extract and the commercial noni juice (the latter used in the vast majority of experiments). This was in part motivated by our hypothesis that the high indentation frequency of *D. sechellia* was because commercial noni juice might be a suboptimal chemical substrate, leading to flies probing the environment more. However, as shown in Supplementary Figure S9b-c, flies displayed similar oviposition behaviors on substrates containing commercial noni juice and fresh noni extract; if anything, commercial noni juice was more attractive, supporting the use of this more standardized chemical stimulus in our work.

2. We now explicitly show the experiment demonstrating that *D. melanogaster* that does not lay eggs does not produce indentations (Figure 2c), suggesting that the higher rate of substrate indentations made by *D. sechellia* is not simply a consequence of lower egg number in this species, but rather reflects its robust sensory probing of the oviposition substrate before proceeding to egg deposition. (This result was initially cited as "unpublished data" in the original manuscript).

3. We now supply all the raw data for the behavioral and new electrophysiology experiments (Source Data files).

Reviewer Figure 1. Comparison of oviposition responses to hexanoic acid versus water.

Single-fly oviposition assays testing the indicated concentration of hexanoic acid versus H₂O (in instant medium) for the indicated genotypes (full genotypes defined in Fig. 6d). Left: oviposition preference index. No statistical differences were observed across genotypes. NS (not significant) $P > 0.05$ (Wilcoxon tests with Bonferroni correction for multiple comparisons); $n = 38-60$ flies, 2 technical replicates. Right: egg-laying rate. Mean values \pm SEM are shown. Statistical differences between groups are indicated. *** $P < 0.001$; ** $P < 0.01$; * $P < 0.05$ (Kruskal-Wallis rank sum test with Nemenyi post-hoc test).

Reviewer #1 (Remarks to the Author):

The authors have satisfyingly addressed the reviewers' points of concern, and I recommend acceptance of the manuscript.

Reviewer #2 (Remarks to the Author):

The authors addressed the technical issues I raised very well. While I believe my comments about the limited conceptual advance this MS provides remain valid, I now support the publication of this MS as it can be considered foundational work on *sechellia* oviposition